# Cancer cells dying from ferroptosis impede dendritic cell-mediated anti-tumor immunity

Bartosz Wiernicki[1,2,3], Sophia Maschalidi [2,4,8], Jonathan Pinney[5], Sandy Adjemian[1,2,3], Tom Vanden Berghe[1,2,3,5], Kodi S. Ravichandran[1,2,4,6] & Peter Vandenabeele [1,2,3,7,8 ✉]

Immunogenic cell death significantly contributes to the success of anti-cancer therapies, but immunogenicity of different cell death modalities widely varies. Ferroptosis, a form of cell death that is characterized by iron accumulation and lipid peroxidation, has not yet been fully evaluated from this perspective. Here we present an inducible model of ferroptosis, distinguishing three phases in the process—'initial' associated with lipid peroxidation, 'intermediate' correlated with ATP release and 'terminal' recognized by HMGB1 release and loss of plasma membrane integrity—that serves as tool to study immune cell responses to ferroptotic cancer cells. Co-culturing ferroptotic cancer cells with dendritic cells (DC), reveals that 'initial' ferroptotic cells decrease maturation of DC, are poorly engulfed, and dampen antigen cross-presentation. DC loaded with ferroptotic, in contrast to necroptotic, cancer cells fail to protect against tumor growth. Adding ferroptotic cancer cells to immunogenic apoptotic cells dramatically reduces their prophylactic vaccination potential. Our study thus shows that ferroptosis negatively impacts antigen presenting cells and hence the adaptive immune response, which might hinder therapeutic applications of ferroptosis induction.

[1] VIB-UGent Center for Inflammation Research, Ghent, Belgium. [2] Department of Biomedical Molecular Biology, Ghent University, Ghent, Belgium. [3] Cancer Research Institute Ghent (CRIG), Ghent University, Ghent, Belgium. [4] Department of Microbiology, Immunology, and Cancer Biology, University of Virginia, Charlottesville, VA, USA. [5] Pathophysiology lab, Department of Biomedical Sciences, University of Antwerp, Antwerp, Belgium. [6] Division of Immunobiology, Department of Pathology and Immunology, Washington University School of Medicine, St. Louis, MO, USA. [7] Methusalem program, Ghent University, Ghent, Belgium. [8] These authors jointly supervised this work: Sophia Maschalidi, Peter Vandenabeele. ✉email: Peter.Vandenabeele@irc.vib-ugent.be

Ferroptosis was coined in 2012[1] and refers to an iron-dependent type of necrotic cell death, characterized by a disruption of the intracellular redox balance and excessive lipid peroxidation[2]. Molecules that modulate iron metabolism and lipid peroxidation can induce or inhibit ferroptosis, such as glutathione peroxidase 4 (GPX4)[3], the glutamate/cystine anti-porter System $X_c^{-}$ [1,4], ferroptosis suppressor protein 1 (FSP1)[5–7], nuclear factor erythroid 2-related factor 2 (NRF2)[8], or p53[9–11]. Recently, ferroptosis has emerged as a powerful tool in anti-cancer therapy to bypass chemotherapy resistance[12,13] and to eliminate metastatic tumors[14]. Furthermore, newly developed ferroptosis inducers have been combined with nanotechnology-based targeting of tumor cells[15,16].

However, despite the promising therapeutic applications of ferroptosis induction, it remains unclear how ferroptosis interacts with the immune system. Therefore, there is a growing need to understand how this newly discovered form of cell death relates to cancer, including immunotherapy. During the last decade, the concept of immunogenic cell death (ICD) of cancer cells has emerged as a type of cellular demise that results in mounting a cytotoxic T-cell (CTL) response targeting cancer cells and con-tributing to tumor eradication. Induction of immunogenic cell death is a resultant of three components in the interaction between cancer cells and the immune system, namely, delivery, processing, and presentation of tumor-associated antigens (TAA) on the surface of dendritic cells (DC), the release of damage-associated molecular patterns (DAMP) propagating adjuvanticity, and chemokine, cytokine, and interferon-driven immune stimulation[17].

To evaluate the role of ferroptosis in inducing ICD, we examine the capacity of ferroptotic cells to modulate the specific anti-tumor response using several settings of prophylactic and ther-apeutic vaccination in vivo, at the level of maturation and pha-gocytosis by DC, and DC's ability to cross-present antigens and activate antigen-specific T cells. Our results provide insight into how ferroptosis, despite its ability to overcome cancer cell death resistance, may interfere with existing cancer (immuno)therapies.

## Results

### Ferroptotic, in contrast to apoptotic and necrotic, cancer cells completely fail to elicit immune protection despite the release of DAMP and cytokines.
Prophylactic cancer vaccination models are a powerful approach to test the immunogenicity of a parti-cular cell death modality[18]. To this end dying tumor cells are initially administered on one flank, and protection against tumorigenesis is tested by subsequent subcutaneous injection of live cancer cells on the other flank (Fig. 1a). Treatment of fibrosarcoma MCA205 cells with ML162, a class II ferroptosis inducer that directly inhibit GPX4[19], leads to ferroptosis char-acterized by increased levels of lipid peroxidation. This cell death is inhibited by the iron chelator Deferoxamine (DFO) and the lipid peroxidation inhibitor Ferrostatin-1 (Fer1), but not by apoptosis (zVAD-fmk) and necroptosis (Necrostatin 1s, Nec1s) inhibitors (Supplementary Fig. 1a–c). Prophylactic vaccination with ML162-killed ferroptotic MCA205 cancer cells (Supple-mentary Fig. 1d) does not protect against a subsequent challenge with live tumor cells, in contrast to previously described vacci-nation with MCA205 cancer killed by immunogenic apoptosis[20] (Fig. 1a). No difference in the tumor growth rate is observed among the animals that succumbed to the challenge regardless of the vaccination (Supplementary Fig. 1e).

The immunogenicity of dying cancer cells depends on the release of DAMP, chemokines, cytokines, and type I interferons (IFN)[17,21]. Analysis of supernatants from MCA205 cancer cells stimulated with ML162 and two other ferroptosis inducers, RSL-3 (class II) and Erastin[22] (class I, relying on the blockage of the cystine–glutamate antiporter system $X_c^{-}$) (Supplementary Fig. 1a–c) reveal the presence of DAMP including ATP and HMGB1 as well as cytokines including CXCL1, TNF, and IFN-β (Fig. 1b, Supplementary Fig. 2a). The release of ATP already occurs before cell membrane permeabilization, while HMGB1 detection is revealed concomitantly with plasma membrane rupture (Supplementary Fig. 1f). Exposure of calreticulin, an endoplasmic reticulum protein that is translocated to the plasma membrane surface and facilitates phagocytosis in ICD[20,23], can be detected for ML162, RSL-3 and Erastin. These induce eventually similar maximal CRT levels per cell as those observed during immunogenic apoptosis by doxorubicin (Fig. 1c, Supplementary Fig. 2b, c). However, contrary to doxorubicin treatment, calreticulin exposure in ferroptosis occurs in small population of non-permeabilized cells at the later stages when the majority of cells are already dead and does not reach statistical significance compared to the untreated cells (Fig. 1d, Supplementary Fig. 2c). Altogether, these data show that ferroptosis death despite the release and exposure of immunogenic cell death-related DAMP and cytokines is not an immunogenic type of cell death.

### Synchronizing ferroptosis induction allows the distinction of three non-immunogenic phases.
Because we observed the release of ATP, CXCL1, and IFN-β already at the early stages of cell death, we decided to perform a prophylactic vaccination assay using partially killed MCA205 cells. Like previously reported[24], a timely treatment of MCA205 cells with GPX4 inhibitor for 6 h does not result in complete cell death. Indeed, the prophylactic vaccination with these incompletely induced ferroptotic cells resulted in tumor growth at the vaccination site (Supplementary Fig. 3a) making the interpretation of the vaccination data pro-blematic, according to the gold standard of prophylactic cancer vaccination[18]. The addition of live cells during prophylactic vaccination and the consecutive tumor growth may lead to the development of immunogenicity against the challenge through so-called concomitant immunity[25]. The phenomenon has been described in melanoma tumors[26] and occurs also in MCA205 model (Supplementary Fig. 3b, c). To overcome this issue and to better delineate the kinetics of the immunomodulatory factors released by ferroptotic cells, we designed an inducible model of ferroptosis via doxycycline (dox)-inducible knockdown of GPX4 (denoted iGPX4KD) (Fig. 2a). Administration of doxycycline induced cell death in iGPX4KD cells within 48–72 h that was completely rescued by Fer1 (Supplementary Fig. 4a–c). After washing out the Fer1 inhibitor following doxycycline induction (Fig. 2b) synchronized cell death reached nearly 100% at 8 h (Fig. 2c, Supplementary Movie 1, Supplementary Movie 2). The analysis of synchronized cell death in iGPX4KD cells revealed that ferroptosis starts with a strong increase in lipid ROS at the plasma membrane and cytosolic ROS production (coined "initial ferroptosis", 1–2 h), followed next by rounding of the cells and release of ATP and exposure of calreticulin ("intermediate fer-roptosis", 3–4 h) and eventually ending in plasma membrane permeabilization coinciding with the release of LDH, HMGB1, cytokines, chemokines and IFN ("terminal ferroptosis", 5–8 h) (Fig. 2c). The observed cell death induction is not associated with NF-κB activation (Supplementary Fig. 4d) suggesting that the production of cytokines during cell death occurs through another signaling cascade. Similar to drug-induced ferroptosis, the expo-sure of calreticulin occurs in a subpopulation of cells only shortly before their permeabilization and can be associated with reaching late intermediate/terminal stage of ferroptosis (Fig. 2d, e). A point of no return in iGPX4KD cells was determined at 2 h following induction and removal of Fer1. At this time point, re-adding Fer1

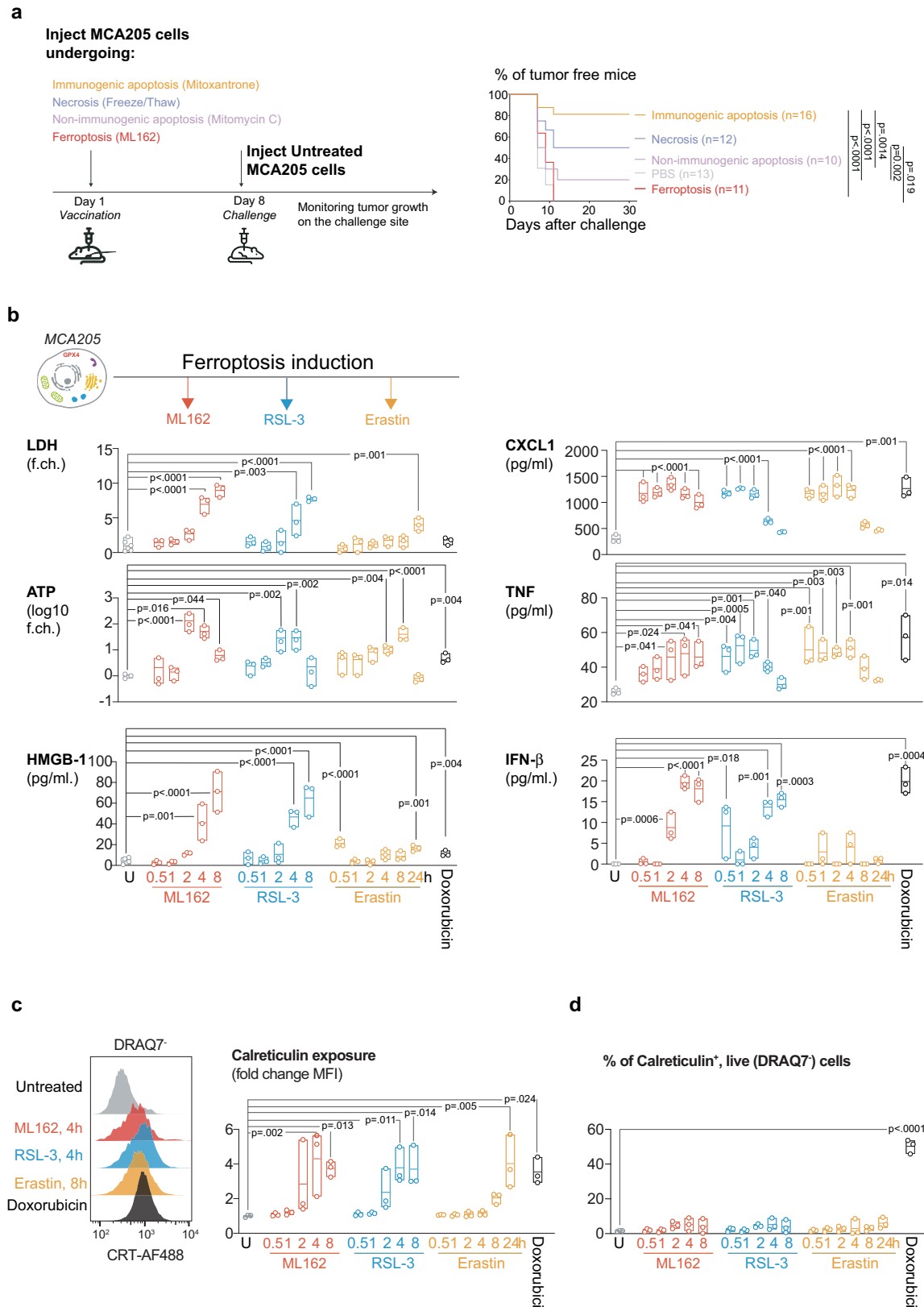

does not prevent further progression to complete cell death (Fig. 2f), while re-adding Fer1 at 1 h following synchronized ferroptosis induction results in complete viability rescue.

Having defined "initial" and "terminal ferroptosis" as well as the time point when cells are still viable, but ferroptosis is inevitable, we could use these conditions in a prophylactic vaccination experiment without the risk of tumor growth on the vaccination site. Prophylactic vaccination with neither initial nor terminal ferroptosis elicited a protective response against a challenge with living tumor cells underlining the absence of immunogenicity of ferroptotic cancer cells following prophylactic vaccination (Fig. 2g). Tumors originating from the challenge had

**Fig. 1 Ferroptosis does not induce immunological protection against cancer cells despite the release of DAMP. a** Prophylactic vaccination model with MCA205 cells was used to assess the immunogenic potential of ferroptosis (ML162, 0.5 μM, 14 h) via their ability to induce protection against cancer tumor growth. Immunogenic (Mitoxantrone, 1 μM, 24 h) and non-immunogenic (Mitomycin C, 30 μM, 24 h) apoptosis, as well as accidental necrosis (Freeze/thaw, three cycles) conditions, were used as controls. Kaplan–Meier curves represent the % of tumor-free mice after the challenge with live cancer cells. Data from $n = 3$ independent experiments was analyzed by Kaplan–Meier simple survival analysis. **b** DAMP release from MCA205 cells stimulated with three inducers of ferroptosis: ML162 (0.5 μM), RSL-3 (0.5 μM) and Erastin (2.0 μM) as well as doxorubicin (1 μM, 24 h). Release of LDH, ATP, HMGB1 as well as cytokines CXCL1, TNF and IFN-β was analyzed at different time points after cell death induction. Data from $n = 3$ biologically independent samples are presented as floating bars with bounds showing the range and the center showing the mean. Data analyzed by one-way ANOVA with Dunnett's post-hoc tests for each ferroptosis stimuli separately and two-sided t-test for doxorubicin treatment comparing the values to the untreated sample. **c** Analysis of calreticulin exposure on the surface of ferroptotic cells, ML162 (0.5 μM), RSL-3 (0.5 μM) and Erastin (2.0 μM). Treatment with doxorubicin (1 μM, 24 h) served as a positive control. Histograms represent the fluorescence intensity detected in non-permeabilized cells. Bounds of the bars show the range, and the center shows the mean of fold change in fluorescence intensity generated by comparing the MFI of treated cells to MFI of untreated cells. Data were generated from $n = 3$ biologically independent samples. One-way ANOVA, with Dunnett's post-hoc test for each ferroptosis stimuli and two-sided t-test for doxorubicin treatment comparing the obtained values to the untreated sample **d**. The % of live cells exposing calreticulin during cell death. Data presented as range (box bounds) and mean (center) of $n = 3$ independent experiments. One-way ANOVA with Dunnett's post-hoc test for each ferroptosis stimuli and the t-test for the doxorubicin treatment in comparison to the untreated sample.

a similar dynamic of growth regardless of the stage of cell death during vaccination (Fig. 2h). Our data demonstrate that both initial and terminal ferroptotic cancer cells are unable to generate anti-cancer protection following prophylactic vaccination, despite the release of DAMP, cytokines, chemokines, and IFN.

**Initial ferroptotic cancer cells reduce DC maturation and are not engulfed.** To unravel how ferroptotic cancer cells—despite releasing DAMP, cytokines, chemokines and IFN—are not immunogenic, we performed a set of experiments to study the interaction between ferroptotic dying cancer cells and DC. This interplay is crucial for ICD induction[18] and involves phagocytosis of dead cancer cell corpses by DC as well as maturation and cytokine production required for clonal expansion of TAA-specific cytotoxic T cells. Co-incubation of bone marrow-derived dendritic cells (BMDC) with ML162, RSL-3 and Erastin-induced ferroptotic cancer cells cause increased exposure of CD86, CD40, and MHCII^high (Supplementary Fig. 5a) indicative for strong DC maturation. Further experiments relying on the synchronized iGPX4KD model of ferroptosis revealed how each stage of the ferroptosis process impacted the maturation of BMDC (Fig. 3a). We found that co-incubation with the initial stage of ferroptotic cancer cells negatively impacts the level of BMDC maturation. In contrast, when dendritic cells are co-incubated with ferroptotic cancer cells at the intermediate and late stage, they mature to a much greater extent as revealed by increased expression levels of CD86, CD40, and MHCII^high, while the expression of PD-L1 decreased (Fig. 3b). Interestingly, BMDC exposed to synchronized ferroptotic cancer cells (early, intermediate, late) in contrast to UVB-treated apoptotic cells did not produce substantial amounts of cytokines related to inflammation (IL-6, IL-12, TNF, IFN-β) and adaptive immune response (IL-10, IFN-γ) (Fig. 3c). Similar results were obtained with chemically induced ferroptosis (ML162, RSL-3, Erastin) except for IL-6 induction by ML162-killed cells (Supplementary Fig. 5b) suggesting a non-specific effect of the ML162 drug in this context. Altogether, these data reveal the inherent inability of the ferroptotic cancer cells to elicit profound cytokine and interferon production in BMDC in contrast to the same UVB-treated cancer cells, as a prototype of apoptotic ICD (Fig. 3c).

Unlike apoptosis[27,28] or necroptosis[29,30], ferroptosis does not evoke the exposure of phosphatidylserine at the outer leaflet of the plasma membrane prior to cell membrane permeabilization, serving as an "eat me" signal during efferocytosis by phagocytes (Fig. 3d, Supplementary Fig. 6a). While the products of LPO in the membrane have been suggested as ligands for DC in the so-called lipid whisker model during recognition and phagocytosis

by CD36[31,32], incubation of BMDC with synchronized iGPX4KD cells at the initial and terminal stage of ferroptosis results in the engulfment only of the latter stage (Fig. 3e), similarly to chemically induced ferroptotic cells (Supplementary Fig. 6b). Using macrophages as phagocytes confirmed the delay in uptake of ferroptotic cells (Supplementary Fig. 6c). Note, that the observed phagocytosis is independent of PS (Fig. 3f) unlike the uptake of apoptotic cells[28] (Supplementary Fig. 6d) and is also not CRT-dependent (Fig. 3g). The accumulation of lipid droplets has been associated with impaired functionality of dendritic cells in terms of their antigen cross-presentation capabilities[33]. The analysis of BODIPY 493/503 probe fluorescence intensity in the dendritic cells incubated with MCA205 cells showed that co-culture with ferroptotic cancer cells increases the levels of lipid droplets in BMDC (Fig. 3h). Further experiments using flow cytometry confirmed that incubation with ferroptotic cells increases BODIPY 493/503 fluorescence in BMDC and the exposure to the initial ferroptosis is the most significant (Fig. 3i). The analysis of dendritic cells co-cultured with MCA205 killed by drug-induced ferroptosis, apoptosis or by accidental necrosis showed that exposure to ferroptotic corpses causes stronger lipid droplet accumulation in BMDC than apoptotic cells but is on the same level as accidental necrosis cells (Supplementary Fig. 5c).

**Engulfment of ferroptotic cells by DC suppresses the expression of genes associated with adaptive immune response.** To further explore the effect of ferroptotic cancer cells on dendritic cells, we performed transcriptomic analysis of BMDC engulfing ferroptotic corpses. Fluorescently labeled human ferroptotic Jurkat cells killed by ML162 were co-cultured with mouse BMDC for 4 h. Sorted BMDC with ferroptotic cargo (BMDC-*fer*) were subjected to murine transcriptome analysis using untreated BMDC as a control (Fig. 4a). Among the differentially regulated genes (Fig. 4a), we found those directly impacting DC function in generating adaptive immune response (Fig. 4b). We observed downregulation of members of the NF-κB family, specifically *RelB*, *c-Rel* and *NFKB1* possibly explaining the low levels of cytokine production in BMDC exposed to ferroptotic cells. We also observed downregulation of the *Jak2*, *Stat4* signaling molecules, which may affect the autocrine effect of IL-12 during DC priming[34]. Interesting patterns were also observed in genes associated with chemotaxis. BMDC engulfing ferroptotic cells had decreased expression of chemokine *Ccr6*, and *Ccr7*, the latter being required for the trafficking of the DC to the lymph nodes for antigen presentation[35]. At the same time, there was an upregulation of genes coding chemokines, e.g., *Ccl3*, *Ccl4*

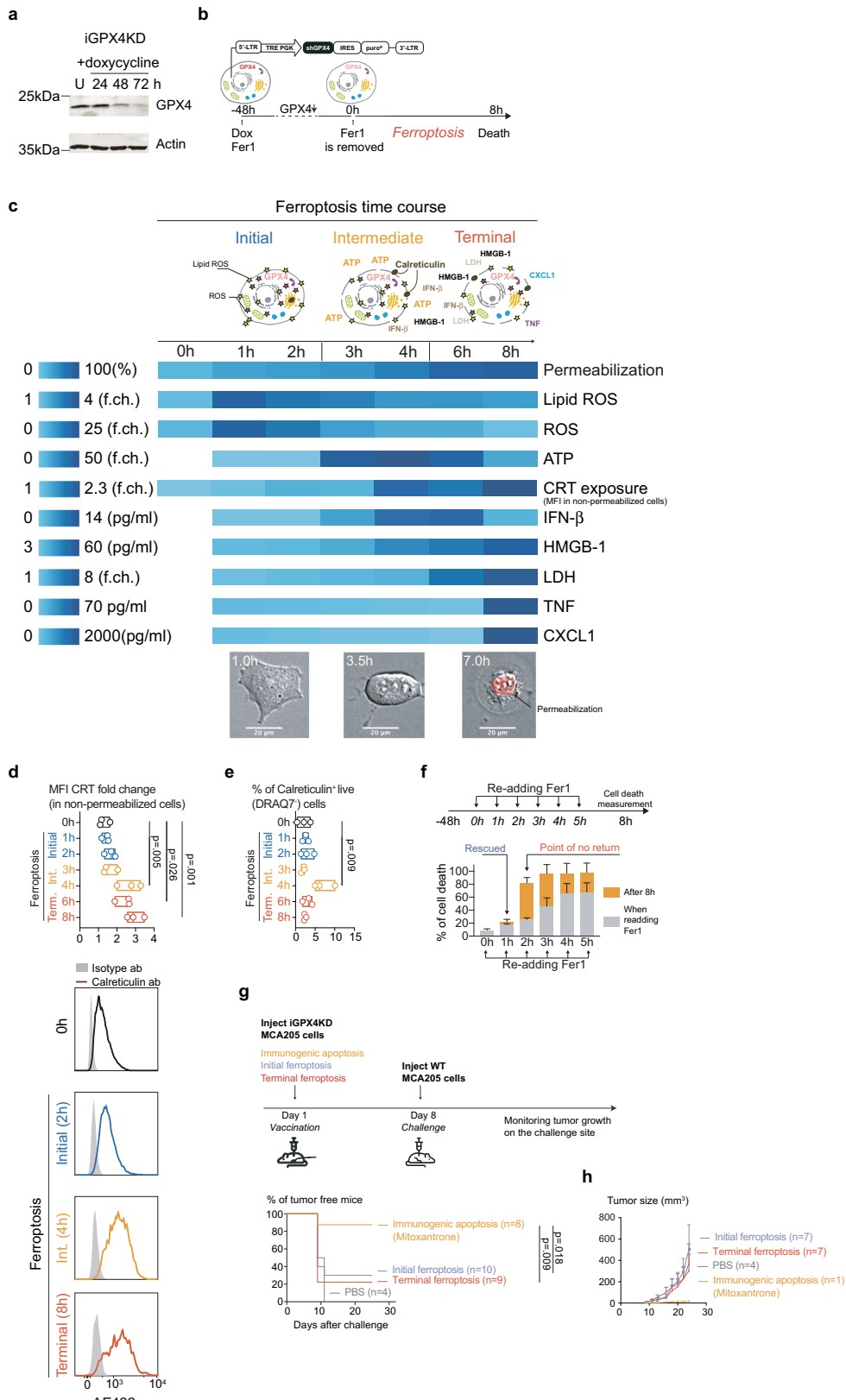

previously linked to the attraction of T cells to the tumor bed[36]. BMDC carrying a ferroptotic cargo strongly upregulated the expression of IFN-β and TNF, both of which positively correlate with the induction of immunogenicity[37,38]. An unbiased global pathway analysis (GSEA) revealed a picture that ferroptotic cells negatively regulated genes involved in the induction of T-cell proliferation, differentiation, and inflammatory response (Fig. 4c) while at the same time upregulated programs related to migration, due to the strong increase in chemokine gene signature. These data suggest that exposure to ferroptotic cancer cells negatively impacts the antigen-presenting features of DC at a transcriptional level.

**Fig. 2 Ferroptotic cells are not immunogenic regardless of the stage of cell death. a** Induction of GPX4 knockdown by doxycycline administration (1 μg/ml) in iGPX4KD MCA205 cell line measured by western blotting. One of two independent experiments is presented. **b** Scheme of ferroptosis induction in iGPX4KD cells. **c** Analysis of lipid ROS, cytosolic ROS accumulation in the cells, levels of calreticulin on the surface of dying, non-permeabilized cells, and ATP, HMGB1, LDH, IFN-β, TNFα, and CXCL release from cells during ferroptosis. Three stages of ferroptosis can be distinguished: initial ferroptosis where cells experience accumulation of lipid ROS; intermediate ferroptosis involving partial permeabilization and release of ATP and exposure of calreticulin, and terminal ferroptosis with complete permeabilization and release of LDH, HMGB1, and cytokines. Data obtained from $n = 3$ independent samples are presented as mean. Microscopy pictures represent one of $n = 2$ independent live cells imaging experiments. **d** The analysis of calreticulin exposure during ferroptosis. Data generated from $n = 3$ biologically independent samples are presented as floating bars with bounds as the range and center as median of the relative MFI prior to membrane permeabilization; histograms represent the shift of calreticulin fluorescence in non-permeabilized cells. One-way ANOVA with Dunnett's post-hoc test in comparison to the '0 h' sample. **e** The % of live cells exposing calreticulin during the process of ferroptotic cell death. Data generated from $n = 3$ biologically independent samples are presented as floating bars with bounds as the range and center as median of CRT$^+$ cells. One-way ANOVA with Dunnett's post-hoc test in comparison to the '0 h' sample. **f** Establishing the point of no return for ferroptosis in iGPX4KD cells. Re-adding ferroptosis inhibitor Fer1 2 h or later after cell death induction does not rescue cells from dying. Data presented as mean ± SEM from $n = 3$ independent experiments. **g** Prophylactic vaccination model using iGPX4KD cells at the initial and terminal stage of cell death. Mitoxantrone-treated (1 μM, 24 h) iGPX4KD cells undergoing immunogenic apoptosis served as a positive control. Kaplan–Meier curves represent the effectiveness of dying iGPX4KD cells in preventing tumor growth on the challenge site. The experiment was performed $n = 2$ and analyzed by Kaplan–Meier simple survival analysis. **h** The tumor growth on the challenge site in prophylactic vaccination model. Only data presented as mean ± SEM from animals that developed the tumor is shown.

**Ferroptotic cells inhibit cross-presentation of soluble antigens.** To validate the results of RNAseq of BMDC-*fer* cells, we functionally addressed the potential of BMDC to perform antigen cross-presentation following exposure to ferroptotic cancer cells and to induce clonal expansion of antigen-specific CTLs. To this end, we incubated BMDC with soluble antigen ovalbumin (OVA) in the presence of ferroptotic or necrotic cells for 16 h. Next, the supernatant containing non-internalized antigen and dead cell corpses was removed and the BMDC were incubated with OVA-specific naïve CD8$^+$ T cells isolated from OT-I Rag2$^{-/-}$ transgenic mice[39] (Fig. 5a). The exposure of BMDC to chemically induced ferroptotic cancer cells negatively impacted the ability of OVA-loaded BMDC to support the proliferation of OVA-specific CD8$^+$ T cells (Fig. 5b, c). The inhibition of antigen cross-presentation was more pronounced with RSL3- and Erastin-induced cell death, although at a higher ferroptotic cell to BMDC ratio the exposure to ML162-induced ferroptotic cancer cells also resulted in reduced antigen-specific T-cell proliferation. To validate and exclude the interference of drug transfer during antigen cross-presentation, BMDC were exposed to antigen from the initial, intermediate, or terminal stages of ferroptotic cell death. Co-incubation with iGPX4KD cells regardless of the stage of cell death showed similar, diminished proliferation of CTL (Fig. 5d, e) suggesting that the presence of ferroptotic cell corpses rather than the release of soluble molecules negatively impact the proliferation of CD8$^+$ T cells. These data also put forward that the release of pro-immunogenic DAMP like ATP or HMGB1 that occur during ferroptosis is not able to alter this response.

**Ferroptosis is less potent in controlling tumor growth compared to apoptosis and necroptosis and reduces the immunogenicity of apoptosis.** We then compared the immunogenicity of different cell death modalities in a prophylactic vaccination model using OVA-expressing non-tumorigenic BM1 mouse embryonic fibroblasts cells dying by apoptosis, necroptosis, or ferroptosis (Supplementary Fig. 7a). The results showed that apoptotic and necroptotic cells are superior in protecting against challenge with OVA-expressing B16 cancer cells (Fig. 6a) and retarding tumor growth (Fig. 6b) suggesting that ferroptotic in contrast to apoptotic and necroptotic cancer cells impair the processing and presentation of tumor-associated antigen. The inability of ferroptotic cancer cells in mounting an anti-tumor response was also confirmed in the therapeutic vaccination model utilizing conventional dendritic cell type I (cDC1) loaded with a cargo of OVA-expressing ferroptotic cells. As a positive control

for efficient immunogenic cell death, we used the same cancer cells that underwent necroptosis[25,40]. Immunocompetent mice already bearing B16-OVA tumors were injected with $5 \times 10^5$ cDC1 with engulfed BM1-OVA cargo (Fig. 6c, Supplementary Fig. 7b). Follow-up measurements of the tumor showed that cDC1 with a ferroptotic cargo were less potent in inhibiting tumor growth for the following week than the same setting with necroptotic cells (Fig. 6c). Additionally, injection with cDC1 carrying necroptotic cells significantly delayed the time of animal euthanasia due to the size of the tumor compared to cDC1 carrying ferroptotic cells (Fig. 6d). Finally, as in most cancer treatments, a mixture of cell death modalities is observed[41], we examined the potential crosstalk between ferroptosis and immunogenic apoptosis in a population of cancer cells. For this, we performed a prophylactic vaccination experiment using mixes of immunogenic apoptosis and ferroptosis. The addition of ferroptotic cells strongly diminished the immunogenic potential of mitoxantrone-killed cells (Fig. 6e), although, in those animals that succumbed to challenge there was no significant difference in tumor growth (Supplementary Fig. 7c). These data suggest that ferroptosis may possess immunoregulatory properties that can affect the responses of neighboring cancer cells that die in an immunogenic way.

**Discussion**

Ferroptosis is a newly described form of regulated necrotic cell death, characterized by free iron-dependent lipid peroxidation[1]. Preventing ferroptosis is part of cellular homeostasis maintained by a continuous activity of GSH-dependent GPX4 activity[42]. Failing to do so contributes to many pathological conditions such as ischemia/reperfusion injury[43–45], neurodegenerative diseases[46], or diabetes[47]. On the other hand, induction of ferroptosis also plays a crucial role in suppressing carcinogenesis[10] and tumor growth[3,16]. Accumulating evidence suggests that ferroptosis may be utilized in cancer treatment as a novel way to circumvent therapeutic resistance[13], by sensitizing radiotherapy[48,49] and by combination with conventional chemotherapy[50,51]. Ferroptosis induction by blocking GPX4 was particularly potent in experimental models of lung metastasis, because of the oxygen-rich environment favoring lipid peroxidation[52]. Ferroptosis inducers have also been combined with nanotechnology, specifically targeting tumor cells[15,16]. Less is known on the immunoregulatory features of ferroptotic cell death. Ferroptotic cells have been shown to release immunogenic DAMP such as HMGB1[24,53] and to expose calreticulin at the surface[54], and were therefore postulated as an immunogenic cell death modality[24]. In the present

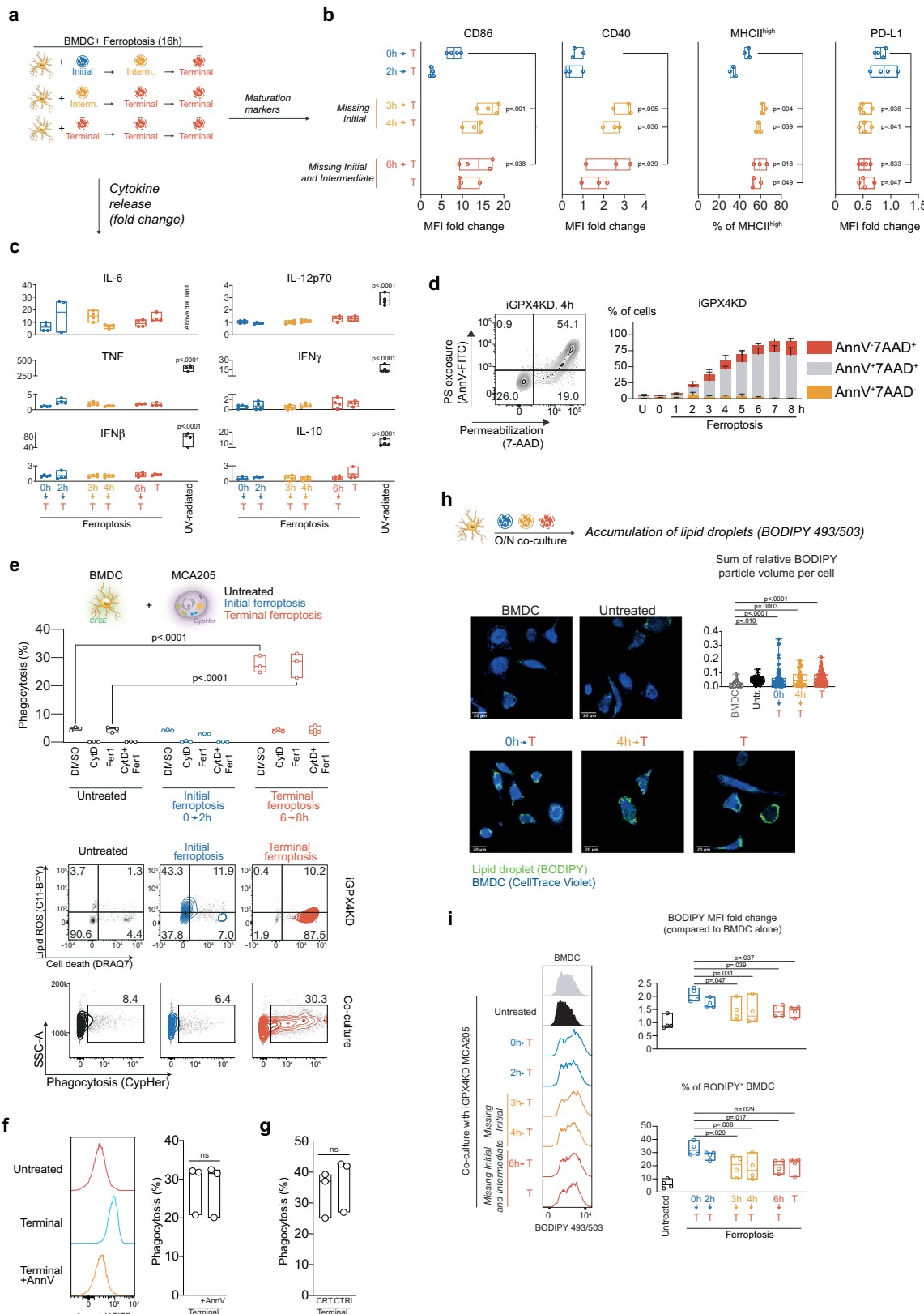

work, we investigated the potential immunogenicity of ferroptotic cancer cells at multiple levels: their efficacy in eliciting a prophylactic cancer vaccination, the release of DAMP, cytokines, chemokines and IFN, and finally its interplay with DC at the level of efferocytosis, transcriptional regulation and antigen cross-presentation.

Adjuvanticity of cell death has been attributed to the concerted action of the release of DAMP, cytokines, chemokines and IFN[17]. In our work, we showed the exposure of calreticulin in a sub-population of ferroptotic cells shortly before their rupture, as well as ATP, HMGB1, CXCL1, TNF and IFN-β release in the course of a drug- and genetically induced models of ferroptosis in cancer

**Fig. 3 Initial ferroptosis impairs the maturation of dendritic cells. a** MCA205 cells with depleted levels of GPX4 and doomed for ferroptosis were co-incubated with dendritic cells for 16 h. To address the role of each ferroptosis stage on the dendritic cell maturation, iGPX4KD cells at different stages of cell death were used. In all conditions iGPX4KD cells reached terminal stage during the co-culture. **b** The analysis of the maturation of dendritic cells incubated with ferroptotic cells. The levels of CD86, CD40, PD-L1 and percentage of dendritic cell population expressing high levels of MHCII was assessed by flow cytometry measurements. Data from $n = 3$ independent samples for CD40 and MHCII measurement and $n = 4$ independent samples for CD86 and PD-L1. Data are presented as floating bar plots with box bounds representing the range and center showing the median of obtained measurements. One-way ANOVA, with Dunnett's post-hoc test analyzing comparison to '0 h' sample. **c** The analysis of cytokine production from the dendritic cells incubated with ferroptotic cells. Data from $n = 4$ independent biological replicates and presented as floating bars with bar limits showing the range and the center describing the median. One-way ANOVA, with Dunnett's post-hoc comparing results from the co-cultures to the untreated bone marrow-derived dendritic cells (BMDC). **d** The analysis of phosphatidylserine exposure on the surface of the ferroptotic cells at different stages of cell death. Representative contour plot of flow cytometry analysis using Annexin V and 7-AAD. Bar graphs show the mean ± SEM of $n = 3$ independent experiments. **e** The level of phagocytosis of untreated and undergoing the initial or terminal ferroptosis. CypHer-labeled MCA205 cells were incubated with CFSE-labeled bone marrow-derived dendritic cells (BMDC) for 2 h. Afterwards the phagocytosis was determined by the detection of CypHer fluorescence in CFSE-stained BMDC. Cytochalasin D and Fer1 were used as inhibitors of phagocytosis and lipid peroxidation respectively. Data from $n = 3$ independent biological samples and is presented as floating bars with bounds representing the range and the center showing the median of % of phagocytic BMDC. Two-way ANOVA with Dunnett's post-hoc test shows the comparison to the untreated condition with the same inhibitor. Contour plots represent the stages of cell death (upper panel) and the gating strategy for phagocytosis detection. **f** The analysis of the dendritic cells phagocytosis of the terminal ferroptotic cells with and without PS blockage by Annexin V. Data presented as median and range and come from $n = 3$ biological replicates, two-sided t-test, ns—not significant. **g** The analysis of dendritic cells phagocytosis of terminal ferroptotic cells with blocked calreticulin. Data presented as floating bars with bounds representing the range and the center showing the median of $n = 3$ biological replicates, two-sided t-test, ns—not significant. **h** The microscopy analysis of lipid droplets accumulation using BODIPY 493/503 nm probe. Bone marrow-derived dendritic cells (BMDC) were incubated with iGPX4KD cells at different stages of cell death O/N. Afterwards, cells were fixed and visualized on confocal microscope. Box plot shows the analysis of the relative volume of detected lipid droplets in the BMDC for each condition and is presented as a box showing median (center line), 25th and 75th percentile (box bounds) and range of the observed results (whiskers) from $n = 3$ independent experiments, where each dot represents one cell in the analyzed images. One-way ANOVA with Dunnett's post-hoc test analyzing the comparison to the untreated BMDC. **i** The flow cytometry analysis of BODIPY 493/503 nm accumulation in the BMDC after the incubation with untreated or ferroptotic iGPX4KD cells. Data presented as floating bar plots with bounds representing the range and the center showing the median of values generated from $n = 4$ independent experiments. One-way ANOVA with Dunnett's post-hoc test analyzing the comparison of ferroptosis conditions to '0 h' condition.

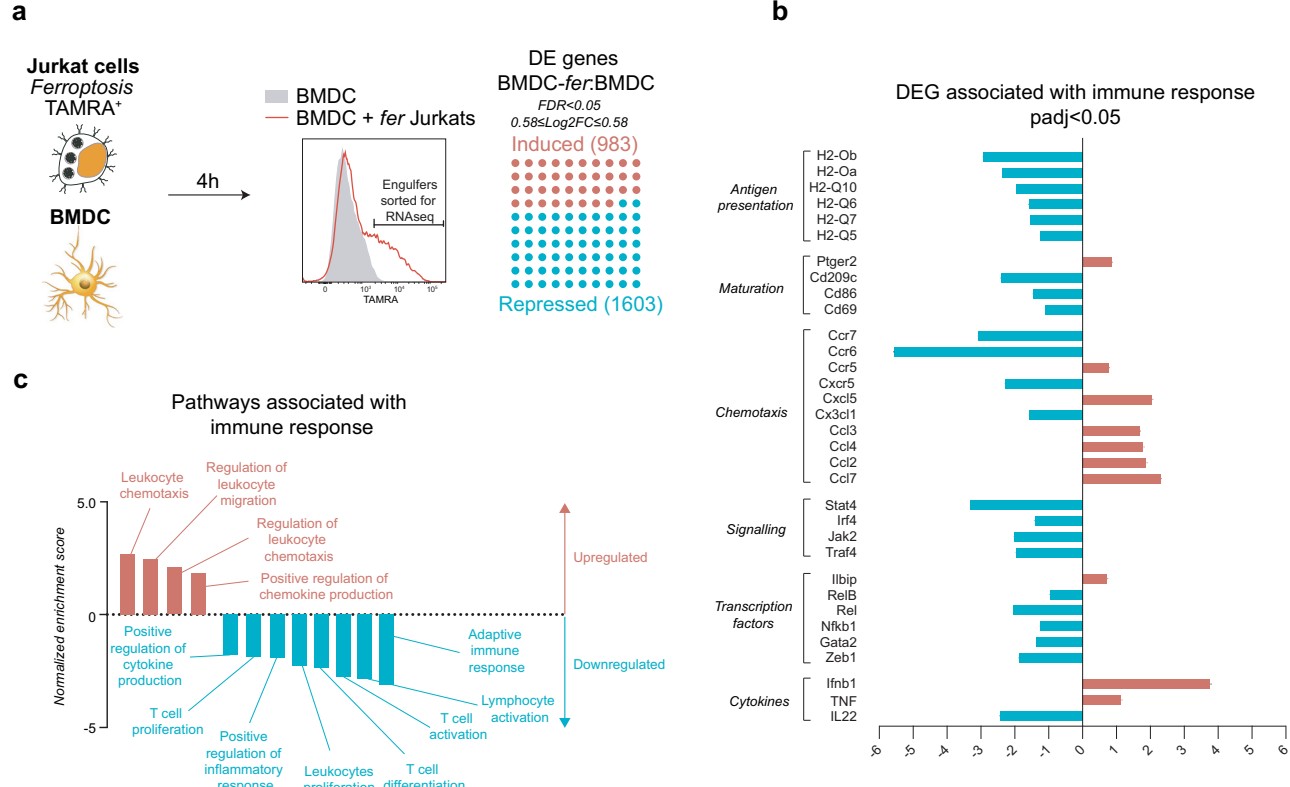

**Fig. 4 Engulfment of ferroptotic cells by DC suppresses expression of genes associated with adaptive immune response. a** Fluorescently labeled ferroptotic Jurkat cells were co-cultured with BMDC for 4 h after which BMDC carrying ferroptotic cargo were sorted and subjected to murine total RNA sequencing ($n = 4$ independently collected samples). Uptake of ferroptotic Jurkat cells led to transcriptional changes in 2586 genes. **b** Expression of selected genes involved in adaptive immune response. **c** GSEA pathway analysis of BMDC carrying ferroptotic cargo revealing transcriptional changes in pathways involved in inducing adaptive immune response.

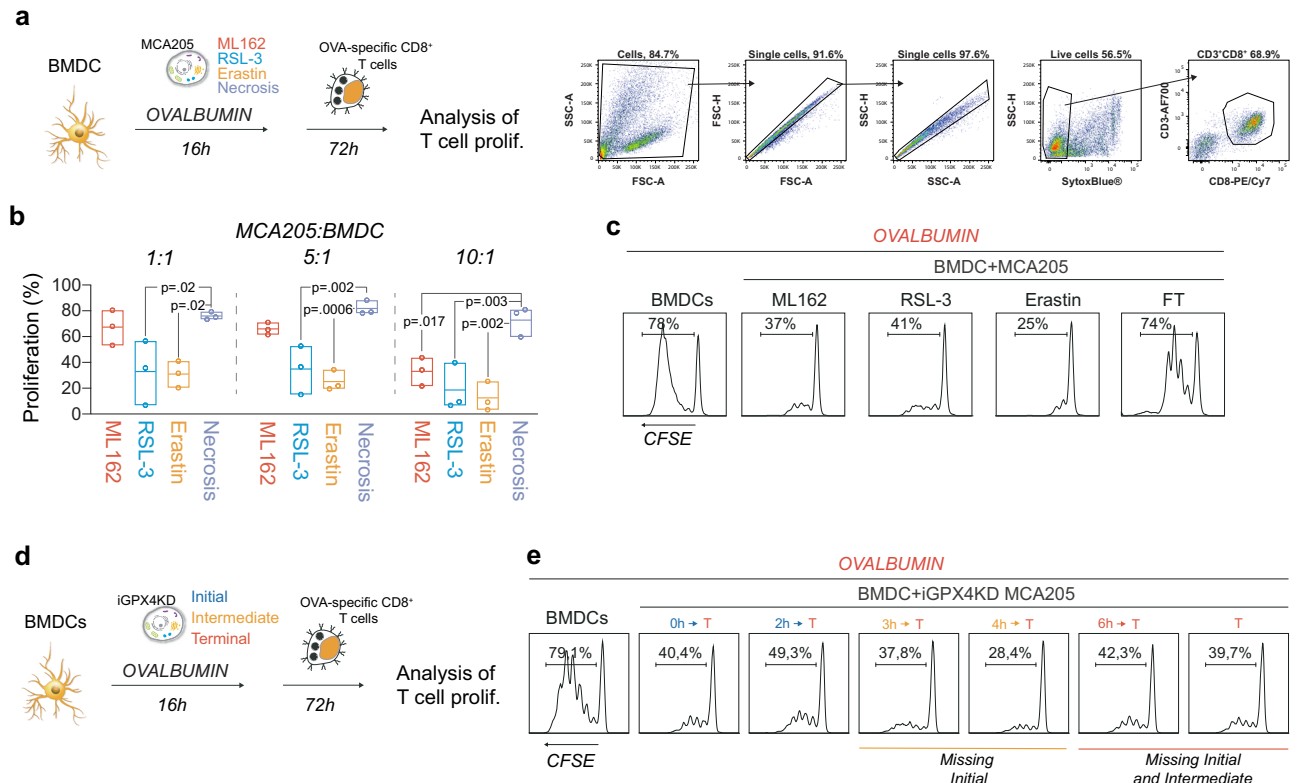

**Fig. 5 Ferroptotic cells impair dendritic cells ability to perform antigen cross-presentation. a** Bone marrow-derived dendritic cells (BMDC) were incubated with soluble OVA in the presence of ferroptotic (ML162 0.5 µM, 14 h; RSL-3, 0.5 µM, 14 h; Erastin 2.0 µM, 24 h) or necrotic (3 cycles of freeze/thaw, FT) cancer cells. Afterwards, fluorescently labeled OVA-specific CD8[+] cells, were added to the co-culture and their proliferation was assessed 72 h later. The flow cytometry dot plots present gating strategy. **b** Percentage of proliferating OVA-specific CD8[+] T cells after co-incubation of ferroptotic and necrotic cells with BMDC. Data comes from $n = 3$ independent experiments and is presented as floating bars showing the range (box bounds) and mean (center line). One-way ANOVA, with Dunnett's post-hoc test analyzing the comparison to the FT condition. **c** Representative histogram of OVA-specific T-cell proliferation after incubation with BMDC exposed to MCA205 cells, ferroptotic or killed by accidental necrosis (FT—freeze/thaw). **d** Scheme of co-culture experiments involving iGPX4KD cell line to address the importance of early events of ferroptosis in inhibiting T-cell proliferation. **e** Representative histograms from $n = 2$ independent experiments of cytotoxic T-cell proliferation after co-incubation with bone marrow-derived dendritic cells with ferroptotic cells at different stages of cell death.

cells. Altogether, the release of DAMP and cytokines during ferroptosis looks very similar to previously described modes of ICD such as apoptosis[55] and necroptosis[25,56]. Based on the genetically induced model of synchronized ferroptosis induction we show that the release of ATP from ferroptotic cells occurs before cell membrane breakage, much like in immunogenic apoptosis where it is facilitated through pannexin-1 channel opening[57]. HMGB1 release during ferroptosis has been reported to contribute to macrophage activation[58], while CXCL1, TNF and IFN-β release contribute to the features of ICD[55,59,60]. On the other hand, calreticulin which has been described as a crucial component of ICD induction[20] and efferocytosis[23], has been exposed on the subpopulation of ferroptotic cells shortly before cell membrane rupture contrary to immunogenic apoptosis, where it was observed with a large time gap prior to cell membrane permeabilization[61]. This difference in kinetics of calreticulin exposure may be because dying ferroptotic cells rapidly proceed to plasma membrane permeabilization, while in ferroptotic cells that die later accumulation of cellular stress such as ER stress can occur which may initiate calreticulin exposure. Nevertheless, the presence of calreticulin on the surface of ferroptotic cells does not facilitate their uptake by the dendritic cells, possibly contributing to the lack of immunogenicity[62]. It is important to underline that while the DAMP release is required for ICD, it does not automatically guarantee the induction of immunogenicity[18]. Several mechanisms that could negatively

impact immunogenicity have been associated with lipid metabolism which could also occur during ferroptosis. Increased expression of cyclooxygenase 2, an enzyme responsible for prostaglandin E2 (PGE2) production, has been marked as a hallmark of ferroptosis[3]. PGE2 can inhibit the activation of the immune system[63,64] and induce FOXP3 transcription factor in T lymphocytes suggesting its positive role in inducing regulatory phenotype[65]. Furthermore, ferroptosis is accompanied by excessive production of oxidized phospholipids[66] and these have been shown to possess strong immunosuppressive properties on APC[67,68]. In line with these potential immunosuppressive mechanisms of ferroptosis, our results show that exposure to co-stimulatory molecules such as CD86 and MHCII on the surface of BMDC decreased when exposed to early ferroptotic cells. The latter is characterized by a higher accumulation of lipids ROS than terminal ferroptotic cells.

However, the crucial feature of immunogenic cell death is the elicitation of properly processed and presented tumor antigens[69]. In that context, we have shown previously that necroptotic dying cancer cells were able to stimulate a broader spectrum of antigen-specific T cells[56]. Here, we document the decreased potential of BMDC to induce antigen-specific T-cell proliferation when exposed to ferroptotic cancer cells compared to untreated cells or cells killed by accidental necrosis. Because dendritic cells underwent maturation (increased CD86, CD40, MHCII[high]), we suspect that the reason may rather lay in a direct effect resulting in

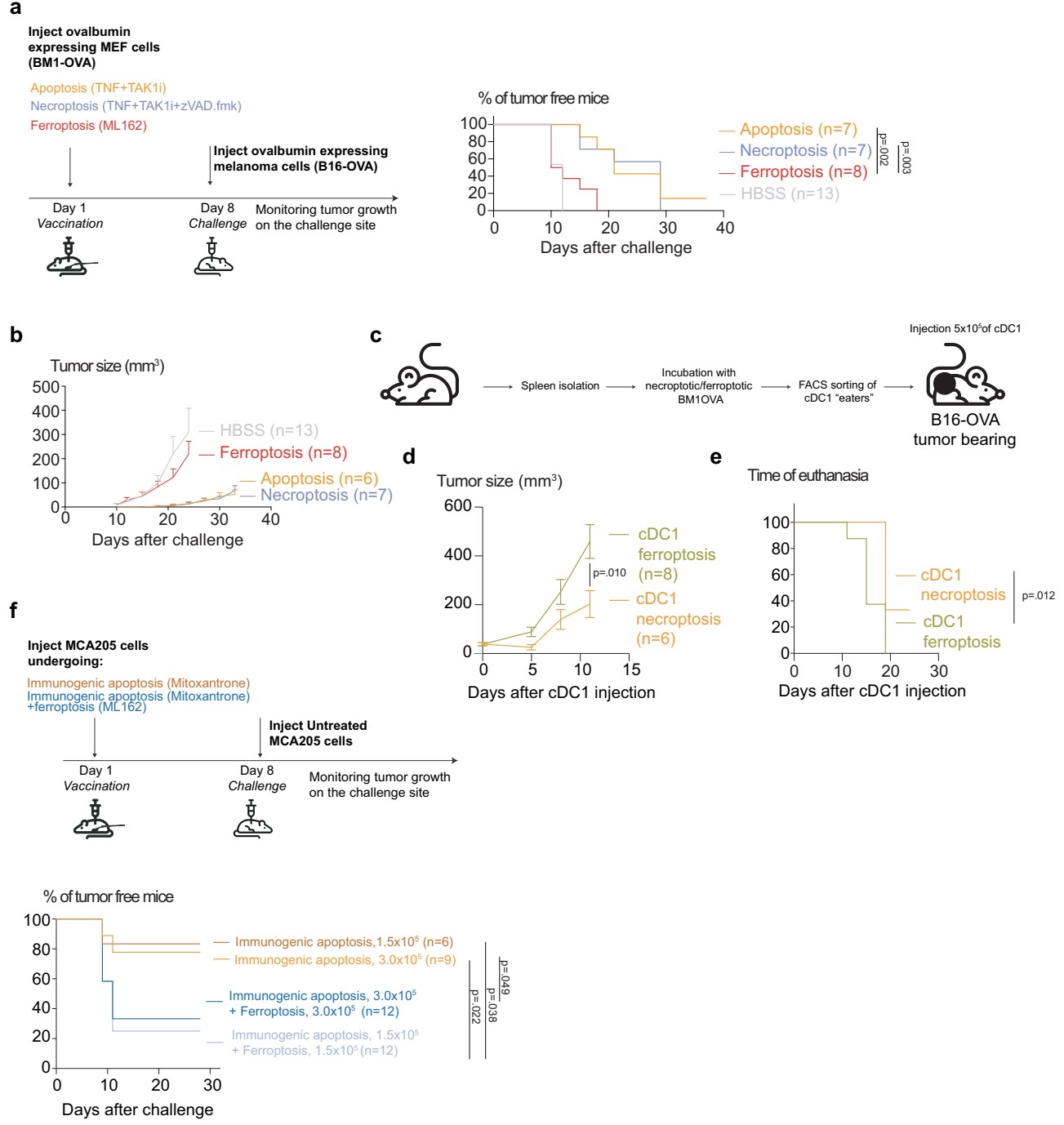

**Fig. 6 Ferroptosis is less potent in controlling the tumor growth compared to apoptosis and necroptosis and diminishes the immunogenicity of apoptosis. a** Prophylactic vaccination model assessing the immunogenicity of ferroptosis. ML162-induced ferroptosis (5 µM, 14 h), in comparison with apoptosis (1000 IU/ml TNF + 10 µM TAK1i, 14 h) and necroptosis (1000 IU/ml TNF + 10 µM TAK1i + 10 µM zVAD.fmk, 24 h). OVA-expressing non-tumorigenic MEF cells (BM1-OVA) were used as a vaccine and live ovalbumin expressing melanoma cells (B16-OVA) were used as challenge. Kaplan–Meier curves represent the effectiveness of ferroptotic cells in preventing the tumor growth at the challenge site. Data were analyzed by Kaplan–Meier simple survival analysis. **b** Tumor size of B16-OVA derived melanoma after vaccination with BM1-OVA cells. Data presented as mean ± SEM. **c** Scheme of the therapeutic vaccination experiment. Conventional dendritic cells type 1 (cDC1) carrying ferroptotic or necroptotic cargo were intradermally injected in melanoma tumor-bearing mice. **d** B16-OVA tumor size progression of animals receiving either cDC1 with ferroptotic (5 µM, 14 h) or necroptotic (1000 IU/ml TNF + 10 µM TAK1i + 10 µM zVAD.fmk, 24 h) cargo. Data presented as mean ± SEM. Statistical significance was determined by two-sided t-test on each day of measurement. **e** Comparison of euthanasia time determined by the size of the tumor. Kaplan–Meier curves show the time of euthanasia, Data analyzed by Kaplan–Meier simple survival analysis. **f** Prophylactic vaccination model using either Mitoxantrone-treated MCA205 cells (1 µM, 24 h) or Mitoxantrone-treated MCA205 mixed with ML162-killed cells (0.5 µM, 14 h). Kaplan–Meier curves show the percentage of tumor-free mice after the challenge with live cancer cells. Data were analyzed by Kaplan–Meier simple survival analysis.

reduced antigen processing and presentation by DC exposed to ferroptotic cells. Incubation with the initial, intermediate and terminal ferroptosis led to the accumulation of lipid droplets, a phenomenon previously linked to depleted capacity to cross-present antigens[70,71]. Additionally, oxPL can also impact the activity of CTLs and block their proliferation in in vitro settings[72] and potentially contribute to CD36-mediated survival of regulatory T cells[73].

In a prophylactic vaccination model, ferroptotic cancer cells failed to elicit immunogenic protection against cancer cells regardless of the stage of the cell death process (initial, intermediate, terminal). Recently, Efimova et al. reported that brief incubation of MCA205 cells with RSL-3, so-called early ferroptotic cells, can provide some protection in prophylactic vaccination model in contrast to late ferroptotic cells. The difference was explained by the release of DAMP and cytokines in the former, while absent in the latter. Although this explanation sounds straightforward, a more detailed analysis of the data puts some caution on these conclusions. Short treatment with ferroptosis inducers does not result in complete cell death induction and retains a high yield of living cells. Indeed, as we found and confirmed by the authors[24], removal of the chemical ferroptosis inducers after a short period of incubation does not result in full cell death. The presence of live cells in the vaccination mixture, whether residual after the treatment, or added to the population of the dead cells, results in the tumor growth on the vaccination site making the interpretation of the data impossible. We solved this problem of partial ferroptosis induction by the development of an inducible model of ferroptosis relying on the doxycycline-dependent knockdown of GPX4 allowing synchronized and complete cell death induction and did not result in the production of the tumor on the vaccination site.

Overall, our data demonstrate that ferroptosis of cancer cells is not an immunogenic type of cell death. We identify several mechanistic aspects or combinations thereof that could explain this rather unexpected finding including reduced efferocytosis of initial ferroptotic cells, reduced cross-presentation and additional transcriptional effects in the dendritic cells that would affect T-cell stimulation and proliferation. Furthermore, we found that these immunosuppressive properties of ferroptotic cancer cells can even overrule apoptotic immunogenic cell death. Altogether, our findings have profound implications for experimental and clinical immunotherapy since some cancer treatments, such as radiotherapy, result in mixtures of cell death[41,74,75]. Our data possibly explain the notion that the ferroptosis correlates with poor prognosis of patients suffering from esophageal[76], gastric[77] and renal[78] carcinoma. Finally, our results advance a concept in which immunogenic cell death is not merely the result of exposure to CRT at the plasma membrane and the release of intracellular contents such as DAMP or cytokines[63]. Indeed, despite these all happen during ferroptosis our data rather suggest that following exposure to ferroptotic cell death, specific molecular brakes modulate the subsequent engagement of the immune system. Identifying these will serve as the springboard for novel therapeutic avenues aimed at treating cancer or modulating aberrant immune responses in affected patients.

## Methods

**Cancer cell lines and cell death-inducing reagents**. MCA205 cells were cultured at 37 °C with 5% CO$_2$ in Gibco RPMI media (#52400, Gibco), supplemented with 10% fetal calf serum. Cells were split when reaching 80% of confluence and detached using 0.25% solution of trypsin and EDTA. In the initial experiments, ferroptosis was induced by ML162 (#AOB1514, Aobious Inc., USA), RSL-3 (#S8155, Absource, Germany) and Erastin (#S7242, Absource) at different concentrations. For the subsequent experiments, ML162 and RSL-3 were used at 0.5 μM and Erastin was used at 2.0 μM. Cell death inhibitors zVAD.fmk (#BACE4026865.0005, VWR International, Belgium) and Nec-1s (synthesized by

the Laboratory of Medicinal Chemistry; University of Antwerp, Belgium) were used at 10 μM, Fer1 (#M60042-2s, Xcess Biosciences, USA) was used at 0,5 μM and DFO (#D-9533, Sigma-Aldrich, USA) at 50 μM. For the prophylactic vaccination model, 1 μM Mitoxantrone (#M-6545, Sigma Aldrich) and 30 μM Mitomycin C (#M-0503, Sigma Aldrich) were used (24 h). To induce accidental necrosis, MCA205 cells were subjected to three freeze/thaw (F/T) cycles using dry ice. For cytokine measurements, MCA205 killed by 24-hour incubation with 1 μM doxorubicin (#D-1515, Sigma Aldrich) were used as a positive control[55]. BM1-OVA and B16-OVA cells were cultured in Gibco DMEM medium supplemented with 10% fetal calf serum and maintained similarly to MCA205 cells. Jurkat cells were cultured in Gibco RPMI medium with 10% of fetal calf serum and split twice a week by transferring a portion of cells to a new culturing flask. In BM1-OVA[79], apoptosis was induced by incubating the cells with 1000 IU/ml TNF and 10 μM TAK1i for 14 h; necroptosis was induced by 24 h incubation with TNF, TAK1i and 10 μM zVAD.fmk; ferroptosis was induced by 10 h stimulation with 5 μM ML162. In Jurkat cells, ferroptosis was induced by 0.5 μM ML162 for 14 h.

**Generation of inducible GPX4 knockdown cellular model for ferroptosis**. The knockdown of GPX4 in MCA205 cells (iGPX4KD) was obtained by lentiviral transduction using pLKO.1-puro vector with a cloned sequence of shRNA specific for GPX4. The obtained pool of cells was selected with 3 μM Puromycin (#P7255, Sigma-Aldrich). Next, cells were seeded at 1 cell/well in a 96-well plate in the presence of the selecting antibiotic. Growing clones were screened for ferroptotic cell death induction upon doxycycline administration (#D8991, Sigma Aldrich) and the most potent clone was selected for further experiments. For these experiments, iGPX4KD cells were cultured for 48 hours in the presence of 1 μM doxycycline (#D8991, Sigma Aldrich). In order to synchronize ferroptotic cell death induction, induction of GPX4 knockdown was done in the presence of 0.5 μM Fer1 for 48 h after which, cells were washed three times to remove Fer1 and medium without Fer1 was added.

**Western blotting**. Cells were denatured in Laemmli buffer by boiling for 10 min. Separation of proteins was performed by SDS-PAGE and proteins were transferred to nitrocellulose membrane (Thermo Scientific) with semi-dry blotting. Membrane was blocked using 5% non-fat dry milk solution in TBS buffer with 0.05% Tween20 (TBST). Incubation with primary antibody against GPX4 (rabbit, #ab125066, Abcam, 1:1000), actin (mouse, #69100, clone C4, MP, 1:15000), phospho-IκBα (Ser32/36) (mouse, #9246, Cell Signaling Technology, 1:1000), IκBα (rabbit, #9242, Cell Signaling Technology, 1:1000) was performed O/N at 4 °C in TBST. After extensive washing, the membranes were incubated with HRP-conjugated secondary anti-rabbit (donkey, #NA934, VWR International, 1:5000), anti-mouse (sheep, #NA931, VWR International, 1:5000) for 1 h in RT. Alternatively, tubulin antibody conjugated with HRP (rabbit, #ab21058, Abcam, 1:10000) was used for 1 h in RT. Membranes were developed using Western Lighting Enhanced Chemiluminescence Substrate (Perkin Elmer).

**Cell death measurement**. Cell death in MCA205 stimulated with class I and class II ferroptosis inducers was measured as described before[80]. Briefly, cells were seeded at 10000 cells/well in 96-well adherent plates and the next day they were treated with cell death inducers. All the inhibitors were added 4 hours before cell death induction. Cell death rate was calculated based on the fluorescence intensity of SytoxGreen® (1 μM). Fluorescence was measured by Fluostar Omega (BMG Labtech GmbH). For PS exposure analysis, cells were collected, washed in 1x Annexin Binding Buffer and stained with Annexin V-APC (#BMS306APC/100, eBioscience, 1:100) or Annexin-FITC (#BMS306FI, ThermoFisher, 1:100), 1.25 μM SytoxBlue® (#S11348, Molecular Probes,) and alternatively 1 μg/ml 7-AAD (#A-1310, Molecular Probes) or 3 μM DRAQ7 (#DR71000, Biostatus UK) and cell death was measured with an LSRII flow cytometer and the data were analyzed using FlowJo 10.2 software.

**Mice**. In all the experiments 6–8 weeks old C57BL/6J female mice purchased from Janvier Labs (#C57BL/6JRj) were used. Mice were housed in individually ventilated cages at the VIB-UGent Center for Inflammation Research in a specific pathogen-free animal facility with the temperature at 20–24 °C and 45–65% humidity with 12 h light/dark cycle and food and water available at libitum. All of the experimental setups were approved by the VIB-Ghent University ethical review board and were performed according to the institutional, national and European animal regulations.

**Prophylactic vaccination model**. The prophylactic vaccination model was performed as described before[61]. Each animal received $3 \times 10^5$ of MCA205 cells or $7.5 \times 10^5$ BM1-OVA cells subcutaneously in their left flank, unless stated otherwise. Seven days later, the animals were challenged with $3 \times 10^4$ MCA205, or $2 \times 10^5$ B16-OVA cells subcutaneously injected into the right flank. For the analysis of immunogenicity of the initial and terminal stages of ferroptosis, iGPX4KD were stimulated for cell death for 2 h (to reach the point of no return) or 8 h after which cells were collected, washed in cold PBS and injected into the left flank of the animals ($3 \times 10^5$). The challenge was performed using wild type cells ($3 \times 10^4$) seven days later. The tumor size on the challenge site was measured every 2–3 days

for 30 days using an electronic caliper (RS Components B.V., Brussels, Belgium). Animals with tumors reaching 1 cm³ were euthanized prior to the end of the experiment by cervical dislocation or inhalation of $CO_2$.

**Analysis of lipid ROS and ROS**. Lipid ROS and ROS measurements were performed as described before[16]. Briefly, cells were seeded in 6 well plates and stimulated for ferroptosis, namely, 2 h stimulation with ML162 and RSL-3 and 8 h stimulation with Erastin. For iGPX4KD, lipid ROS and cytosolic ROS were measured in 1–2 h intervals for 8 h. Cells were collected, washed with PBS and resuspended in PBS in the presence of 2 μM C11-BODIPY (#D3861, Molecular Probes, USA) for lipid ROS measurement or 1 μM of DHR123 (#D-1054, Sigma Aldrich) for cytosolic ROS measurement. After a 10 min incubation at 37° C, cells were washed and the fluorescence intensity of oxidized C11-BODIPY or DHR123 was measured using BD LSR II flow cytometer in Fl-1 channel. The extent of cell death was measured by adding 0.5 μM DRAQ7 or 1.25 μM SytoxBlue®. Data were analyzed using FlowJo 10.2 software. Only non-permeabilized cells were used for the analysis.

**ATP, LDH, HMGB1, and cytokine release**. For the analysis of DAMP and cytokine release, MCA205 cells were seeded at $3 \times 10^5$ cells/well in 6-well tissue culture plates in 2 ml of medium. Supernatant from dying cells was collected at indicated time points and stored at −20 °C. ATP release was measured using CellTiter-Glo® Luminescent Cell Viability Assay (#G7570, Promega, USA). LDH release was measured using colorimetric Pierce LDH cytotoxicity assay (#88954, Life Technologies, USA) according to the manufacturer's protocol. Both ATP and LDH results were normalized to values obtained from untreated (live) cells. HMGB1 release was performed using ELISA assay (#ST51011, Tecan, Switzerland) according to the manufacturer's instructions. Mouse cytokines in cell culture supernatants were determined by a magnetic bead-based multiplex assay using Luminex technology (Bio-Rad, Hercules, CA, USA).

**Live cell imaging**. Live cell imaging of iGPX4KD cells was performed as described before[25]. Briefly, cells (untreated or stimulated for ferroptosis by removal of Fer1) have been seeded $10 \times 10^3$ cells/well in 8 well chamber (#80826, iBidi) in 250 μl of medium in the presence of DRAQ7 (3 μM) and the images were acquired on Leica Sp5 AOBS confocal microscope (Leica), using an ×40 HCX PL Apo UV 1.25 na oil objective. Obtained images were further analyzed and extracted using Fiji 1.53 built on ImageJ 2.1.0 software.

**Analysis of calreticulin exposure**. Untreated or dying MCA205 WT and iGPX4KD cells were collected, washed in FACS buffer (3% FCS in PBS) and stained with anti-calreticulin specific antibody (#ab2907, Abcam, 1:300) or an isotype control (PA5-23094, ThermoFisher) for 30 min on ice. Afterwards, cells were washed three times with FACS buffer and incubated with secondary goat anti-rabbit-FITC antibody (#35553, Thermo Fisher Scientific, 1:300) for 30 min on ice. Following three additional washes, cells were analyzed on LSRII Fortessa using BD FACSDIVA 8.0 software, and DRAQ7 was used as a permeabilization marker. Data were analyzed using FlowJo 10.2 software. Only non-permeabilized cells were used for the data analysis.

**Isolation of BMDC**. BMDC were isolated from the tibia and femur as described before[25]. Briefly, the bones were isolated from the 6–8 weeks old female C57BL/6J mice and cells were flushed out with a syringe. Red blood cells were removed by ACK buffer (#10-548E, Biowhittaker Inc, USA). Next, cells were counted and seeded on suspension plates at $2 \times 10^5$ cells/ml in 10 ml of medium in the presence of 20 ng/ml GM-CSF (#130-095-739, Miltenyi Biotechnumber, USA). A fresh portion of GM-CSF-enriched medium was added on day 3 and replaced on day 6. Cells were seeded for experiments on day 9–10. Around 80% of cells stained positive for CD11c marker at that time.

**Phagocytosis assay**. MCA205 cells were detached from the flask and resuspended in an OptiMEM medium containing 1 μM CellTrackerGreen (#C2925, Thermo-Fisher) for 30 min at 37ºC shaking. Stained MCA205 cells were seeded and killed by ferroptosis inducers (ML162, 0.5 μM, 14 h; RSL-3, 0.5 μM, 14 h; Erastin, 2.0 μM, 14 h) or by repeated F/T cycles. On the day of the assay, $4 \times 10^5$ BMDC were seeded in 2 ml of medium in 6-well suspension plates. Untreated or dead cells were collected, counted and added to the BMDC at the appropriate ratio in 2 ml of BMDC medium. Co-culture was incubated for 2 h at 37 °C. After that, BMDC and dead cells were collected and washed in FACS buffer (3% fetal calf serum solution in PBS). Next, samples were stained with CD11c-APC (#117309, BioLegend, 1:200) antibody in the presence of Fc Block antibody (553142, BD Pharmingen, 1:100) for 30 min at 4 °C. After the staining, samples were washed twice in FACS Buffer and resuspended in a solution containing 1.25 μM of the cell death marker SytoxBlue®. Samples were acquired using BD LSRII. CD11c⁺CellTrackerGreen⁺ positive cells were considered as BMDC that engulfed dead cells. For ImageStream experiments, BMDC and MCA205 were stained with 1 μM CellTraceViolet (#C34557, Molecular Probes) and 1 μM TAMRA (#C2211, Invitrogen) respectively, incubated together for 2 h and acquired by AmnisImageStreamˣ Mk II with 40x magnification. Data

were collected using INSPIRE software and further analyzed by IDEAS 6.2 software. For the experiments describing the uptake of ferroptotic cells at different stages of cell death, MCA205 cells were stained with 1 μM CypHer Red (#PA15405, VWR International) and BMDC with 1 μM of CellTraceViolet for 15 min in OptiMEM medium at 37 ºC. Fer1 (0.5 μM), an inhibitor of lipid peroxidation, and cytochalasin D (2 μM, C8273, Sigma Aldrich), an inhibitor of actin polymerization and consequently phagocytosis, were added 30 min before co-culture. Cell-TraceViolet⁺CypHer⁺ cells were considered to be BMDC that engulfed dead cells. For experiments studying the effect of phosphatidylserine or calreticulin exposure on the phagocytosis of MCA205 cells, experiments were performed as described before[81]. Briefly, CFSE-stained iGPX4KD MCA205, either apoptotic (1000 IU/ml TNF + 10 μM TAK1i, 8 h) or terminally ferroptotic cells were incubated with unconjugated Annexin V (100 μg/$2.5 \times 10^5$cells; BioLegends, 640902) in Annexin Binding Buffer for 30 min in room temperature. Afterwards, cells were washed in Annexin Binding Buffer and added to the CellTraceViolet-stained BMDC for 2 h at 37 °C. The role of CRT in phagocytosis of terminal ferroptotic cells was assessed by incubating ferroptotic cells with anti-CRT (1:300) or isotype control antibody (1:300) in FACS buffer for 30 min at 4 °C. CellTraceViolet-stained BMDC were washed twice with FACS buffer and then incubated with CD16/CD32 (1:100) blocking antibody for 15 min. Washed target cells were incubated with BMDC for 2 h a 37 °C in complete medium. The analysis of phagocytosis was performed using LSR II flow cytometer using BD FACSDIVA 8.0 software. Data were analyzed using FlowJo 10.2 software. The population of CellTraceViolet positive cells that were CFSE positive was considered as engulfing BMDC.

**Analysis of BMDC maturation**. BMDC were stained with 1 μM CFSE for 10 min at 37 °C in PBS, washed and seeded at $5 \times 10^4$ cells/well in 96 well suspension plates. Next, BMDC were co-cultured with MCA205 cells in ratios 1:1; 1:5; 1:10 (BMDC:MCA205) or stimulated with 250 ng/ml of LPS (#L-2630, Sigma Aldrich). After O/N co-culture, cells were collected, washed in ice-cold FACS buffer and stained for surface markers: CD11c-BV650 (#117339, BioLegend, 1:200), MHCII-APC/eFluor700 (#47-5321-82, eBioscience, 1:100), CD86-PE/Cy7 (#105116, BioLegend, 1:200), CD80-PE/Cy5 (#104712, BioLegend, 1:200), CD40-APC (#558695, BD Pharmingen, 1:200) and CD274-PE (#12-5982-82, eBioscience, 1:200). The acquisition was performed with a LSRII flow cytometer using BD FACSDIVA 8.0 software and the analysis was performed using FlowJo 10.2 software.

**Analysis of lipid droplets accumulation**. BMDC incubated with ferroptotic cells O/N were collected, washed twice with PBS in the room temperature and incubated with 1 μM BODIPY 493/503 (#D3922, Molecular Probes) for 15 min at 37 °C. After that, cells were washed twice with PBS, resuspended in PBS containing 1 μM DRAQ7 and immediately analyzed by LSR II Fortessa using BD FACSDIVA 8.0 software and analyzed by FlowJo 10.2 software, or imaged by the confocal microscope.

**Antigen-specific T-cell proliferation**. BMDC:MCA205 co-cultures were seeded at $5 \times 10^4$ cells in 96 well plates in 100 μl of medium with appropriate number of MCA205 cells and model antigen 0.25 mg/ml ovalbumin endotoxin-free (#vac-pova-100, Invivogen, France). Co-culture was maintained for 18 h at 37 °C. Afterward, wells were washed three times with medium to remove MCA205 corpses and soluble antigen and BMDC were co-incubated with CD8⁺ T cells isolated from OT-I Rag2⁻/⁻ transgenic mice using MagniSort® Mouse CD8 T-cell Enrichment Kit (#8804-6822-74, eBioscience) pre-stained for 5 min with 1 μM CFSE (#8804-6822-74, Thermo Fisher Scientific, USA) in PBS. Co-culture was incubated for 72 h at 37 ºC. After that, cells were collected and stained with CD3-AF700 (#56-0032-82, clone 17A2, ThermoFisher, 1:200), CD8-PE/Cy7 (#25-0081-82, clone 53-6.7, ThermoFisher, 1:200) and SytoxBlue®. Analysis was performed on BD LSRII using BD FACADiva 8.0 software and further analyzed using FlowJo 10.2 software.

**RNA sequencing of BMDC carrying ferroptotic corpses**. CellTraceViolet-stained BMDC were co-cultured with ferroptotic Jurkat cells (0.5 μM ML162, 14 h) for 4 h, unbound Jurkat cells were removed by washing with PBS and engulfing BMDC were isolated by sorting with BD FACSAriaᵀᴹ III Cell Sorter. Total RNA was extracted, and an mRNA library was prepared using the Illumina Novoseq6000 platform by Novogene. HISAT2 was selected to map the filtered sequenced reads to the reference genome. BAM files containing mapping results were counted using the featureCounts function in the R package Rsubread. Counting was performed using both mouse and human genomes for comparison although downstream analyses were only performed on mouse data. DEG analysis was then performed using DESeq2 considering all genes with FDR ≤ 0.05 and $0.58 \leq Log_2FC \leq -0.58$. All genes that resulted from the analysis were curated using multiple methods, including literature mining and predictive algorithms including Uniport. Functional analysis of genes with FDR ≤ 0.05, regardless of Log2FC, comprised of GO and GSEA (Gene Set Enrichment Analysis analyses. For GSEA, gene sets from MSigDB were used in this assessment which includes curated gene sets (HALL-MARKS), known pathways (KEGG), and gene ontology terms (BIO PROCESS & MOLECULAR FXN).

**Therapeutic vaccination model**. Therapeutic vaccination was performed as described before[82]. Spleen from C57BL/6J mice was isolated, cut into small pieces and incubated with the mix of Liberase™ TM Research Grade (5401127001, Roche) DNAse I (10104159001, Roche) for 30 min at 37 °C according to previously published protocol[83]. Afterward, the single-cell suspension was washed in 1xHBSS (14175-053, Gibco) and red blood cells were removed by 3 min incubation with ACK buffer (A1049201, Gibco). After that, the cell suspension was incubated with TAMRA labeled BM1-OVA cells killed by necroptosis or ferroptosis for 4 h at 37 °C. Following the staining with antibodies for XCR1-BV650 (#148220, BioLegend, 1:200) and CD11c-BV711 (#563048, BD Pharmingen, 1:200) as well as cell death marker eBioscience™ Fixable Viability Dye eFluor™ 506 (1/1000, #65-0866-14, Thermo-Fisher). CD11c+XCR1+TAMRA+ cells were sorted using FACS. The obtained population was centrifuged, resuspended in 1xHBSS and $5 \times 10^5$ cDC1 carrying ferroptotic or necroptotic cells were injected intradermally in C57BL/6J mice bearing B16-OVA subcutaneous tumors. The growth of the tumor was monitored every 3–4 days using an electric caliper. The size of the tumor was determined by the formula: $(3.14 \times Width \times Length \times Depth)/6$.

**Statistics and reproducibility**. All of the statistical analysis was performed using GraphPad Prism v9.0 software. Microsoft Excel v16.5 was used for the initial data transformation and cleaning when necessary.

For Figs. 1b–d, 2d, e, 3b, c, h, i, Supplementary Figs. 2a, 5b, 5c, One-way ANOVA test was applied with Tukey or Dunnett's post-hoc testing as indicated in figure legends.

For Fig. 5b, Supplementary Figs. 5a, 6b a set of One-way ANOVA tests was applied with Tukey post-hoc testing for each ratio.

For Fig. 3f, g, Supplementary Data Figs. 1c, 6d, two-tailed t-test was applied.

For Figs. 1a, 2g, 6a, 6f Supplementary Fig. 3b, Kaplan–Meier simple survival analysis test was applied to calculate the significance of the observed difference in tumor-free mice.

For Fig. 6e Kaplan–Meier test was applied to calculate the significance of the observed difference in time of euthanasia of tumor-bearing mice.

For Fig. 6c, a set of two-tailed t-test for each of the time points was performed to determine the significance of the tumor size.

For Fig. 3e, Supplementary Fig. 6c, a two-way ANOVA test was performed.

For Supplementary Fig. 1f, a simple linear regression with two-tailed Spearman correlation analysis was performed.

In all floating bars plots, the bar represents the range of the results, and the inner line refers to the median or mean, as indicated in the legend.

**Reporting summary**. Further information on research design is available in the Nature Research Reporting Summary linked to this article.

## Data availability

All data generated or analyzed during this study are included in this published article (and its supplementary information files). The RNA sequencing raw data files have been deposited at GEO under the accession code GSE205069. Source data are provided with this paper.

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

## Acknowledgements

Figures were created with Biorender.com. The authors thank VIB Flow Cytometry Core and VIB Imaging Core for their expertise and help to perform the experiments. We thank Prof. Wim Declercq from VIB-UGent for his advice in constructing the MCA205 iGPX4KD, Dr. Wei Xie from VIB-UGent for his advise on the calreticulin measurement protocol, Prof. Caetano Reis e Sousa from the Francis Crick Institute for BM1-OVA cell line, Benjamin Pavie from the Microscopy Core Facility for quantification of lipid droplets and Dr. Adam Wahida for discussion and critical reading. Research in the Vandenabeele group is supported by EOS MODEL-IDI (grant 30826052), EOS INFLADIS (grant 40007512), FWO senior research grants (G.0C76.18N, G.0B71.18N, G.0B96.20N, G.0A9322N), Methusalem (BOF16/MET_V/007), iBOF20/IBF/039 ATLANTIS, Foundation against Cancer (FAF-F/2016/865, F/2020/1505), CRIG and GIGG consortia, and VIB. S.M. received a European Marie Sklodowska-Curie post-doctoral, individual fellowship (800446) from the European Commission, Horizon 2020 Research and Innovation Framework Program. S.A. was the recipient of a post-doctoral fellowship from FWO (Flanders Research Organization). K.S.R. is supported by FWO (Odysseus grant G0F5716N, EOS DECODE 30837538), Special Research Fund UGent (iBOF BOF20/IBF/037), European Research Council (ERC) (grant agreement no. 835243), grants from NHLBI (P01HL120840), NIAID (R01AI159551), NIGMS (R35GM122542), and the Center for Cell Clearance/University of Virginia School of Medicine. T.V.B. is supported by Excellence of Science (MODEL-IDI & CD-INFLADIS), Strategic Basic Research Foundation Flanders (IRONIX, S001522N), Consortium of excellence at University of Antwerp (INFLA-MED), Research Foundation Flanders (FWO G0B7118N & G0C0119N), FWO Kom op tegen Kanker (G049720N), Industrial research Fund from University of Antwerp (IOF), VLIR-UOS (TEAM2018- SEL018), Charcot Foundation, Stichting tegen kanker (FAF-C/2018/1250), Ghent University and VIB. His lab at the Antwerp University is supported by INFLA-MED, FWO Kom op tegen Kanker (G049720N), IOF, TOP-BOF (32254) and FWO (G0C0119N).

## Author contributions

Conceptualization: B.W., S.M., P.V.; Investigation: B.W., S.M., J.P., S.A. Methodology: B.W., S.M., J.P. Funding acquisition: P.V., S.M., K.R. Writing—original draft: B.W., P.V. Writing—review and editing: B.W., S.M., S.A., P.V., T.V.B., K.R.

## Competing interests

The authors declare no competing interests.
