## [Peer Review File · Nature Communications]

Cancer cells dying from ferroptosis impede dendritic cell-mediated anti-tumor immunityREVIEWER COMMENTS

Reviewer #1 (Remarks to the Author):

The manuscript "Cancer cells dying from ferroptosis impede dendritic cell mediated anti-tumor immunity" written by Bartosz Wiernicki and colleagues is an interesting paper that explores the capacity of ferroptosis to induce the so-called immunogenic cell death "ICD". The major claim of the paper is ferroptosis does not induce ICD even though the production of DAMPS, and inflammatory cytokines. This conclusion is well supported by the data presented in the paper. The methodology is correct and well documented. Note that since I am not a statistician I did not evaluate the methods used for statistics.

Strengths include the use an inducible model of ferroptosis relying on the doxycycline dependent knockdown of GPX4 allowing synchronized and complete cell death induction.

The novelty of this study resides in the description of all the mechanisms usually associated with immunogenic cell death and yet ferroptosis does not induce immunogenicity. This suggests that the commonly studied markers associated with ICD (ATP, HMGB1, IFN β ...) may not be systematically correlated with ICD. This work is important since it challenges the mandatory role of DAMPS and other inflammatory cytokines for the induction of ICD.

I have major comments and questions:

Calreticulin (CRT) exposure has been described as an important feature of type I ICD. Why didn't the authors monitor it? CRT should be monitored.

A recent study published in 2020 seems to show contradictory data. Indeed, Iuliia Efimova has published a study entitled "Vaccination with early ferroptotic cancer cells induces efficient antitumor immunity" *J Immunother Cancer*. 2020; 8(2): e001369. In this paper, authors used the same tumor cell line (MCA205) and demonstrated that early ferroptosis induced ICD contrary to late ferroptosis. Bartosz Wiernicki and colleagues discussed these results as "short treatment with ferroptosis inducers does not result in complete cell death induction and retains a high yield of living cells. Indeed, as we found and confirmed by the authors, removal of the chemical ferroptosis inducers after a short period of incubation does not result in full cell death." This reflection is interesting but not necessarily convincing on several aspects.

- Firstly, when we use anti-cancer drugs in patients, death induction is never 100% synchronized and complete but still can favor immunogenicity. Authors should discuss this point
- Secondly, this would imply that the living cells of the MCA205 model would have a vaccination effect. This was described for some tumor lines and published under the concept of concomitant immunity. This concomitant immunity is inhibited by immunosuppressive mechanisms such as regulatory T cells (Alan N. Houghton Group; *J. Exp. Med.* Volume 200, Number 6, 20 September 2004 771-782). However, this control (i.e. living MCA205 cells) is never included in experiments and has to be included in experiments.
- Thirdly, as shown for mitoxantrone, about 50% of tumor cells are still alive (extended data, figure 1d) and yet the vaccine effect exists. Therefore, a question remains with mitoxantrone in authors' setting: is apoptotic cells or living cells inducing this vaccine effect?

Thus, it is crucial to have the vaccine effect of living MCA205 cells to observe (or not) the induction of concomitant immunity. If so, this would imply that DAMPS and other inflammatory cytokines released by dying cells have an adjuvant effect only in the presence of concomitant immunity (induced by living tumor cells), therefore only with immunogenic tumors. It would be interesting to show by diluting living cells with increasing doses of ferroptosis-induced cell death that concomitant immunity is inhibited. The involvement of Treg in this mechanism should be also monitored. This may provide an explanation for the observation that ferroptosis is associated with poor prognosis in some patients (Zhu L, *Cancer Cell Int* (2021) 21(1):124; Zheng Y, *J Cell Mol Med* (2021) 25(6):3080–90; Jiang X, *J Oncol* (2021) 2021:6635526) even though ferroptosis could be correlated with immune infiltrate (Chen Ji. *Front Oncol*. 2021; 11: 700084).

I have minor comments and questions:

- Incubation time with cell death inducers should be clearly indicated for each experiments.

- Rechallenge with only 30.000 MCA205 should be justified. What is the tumor uptake rate with this seemingly low cell count?
- Figure 1b left panel, doxorubicin data are missing. Should be included.
- Figure 2e and fig 6a and 6b: immunogenic apoptosis; necroptosis. No information of ICD inducer (apoptosis and necroptosis) is indicated in the figure legend
- Figure 4b: Does Cxcr5 stand for CXCR5?

Reviewer #2 (Remarks to the Author):

Wiernicki and colleagues report that ferroptotic cancer therapy is not suitable for immunogenic cell death vaccination therapy as it negatively impacts the downstream immune response. It is a finding that depicts the unresolved immunogenic potential of ferroptotic cell death for prophylactic cancer cell vaccination. The paper is overall elegantly written and presented. Authors eloquently analyze the different stages of ferroptosis throughout the manuscript to identify the characteristics that are required for immunogenic cell death. The data is thorough as it not only compares pharmacological inhibition of Gpx4 by ML160 and RSL3 but also takes a genetic approach by inducible knockout of Gpx4. These systems together along with usage of inhibitor of SLC7A11, transporter for cysteine and ultimately glutathione building block strengthens the value of this study. There are some questions that do remain, however, and additional controls are also required. How ferroptosis induces immunogenic cell death profile (i.e. Release of DAMPs, cytokines, chemokines) yet experiences dampening of the adaptive immune response that is necessary for immunological protection is not well defined and insight here would be greatly appreciated. Provided the authors can address some of the questions below, this paper would be of great interest to the readership at Nature Communications.

Specific comments:

1. In figure 3b, authors illustrate expression of DC activation markers upon presentation with cancer cells at different ferroptotic stages, and show that exposure to cells at intermediate/terminal stages facilitate the upregulation of CD86, CD40 and MHC-II but not initial stage of ferroptosis. Although it is unclear whether the magnitudes of expression marker upregulation in intermediate/terminal stages are to the extent of apoptosis or necroptosis cancer cells, it is nonetheless coherent from the subsequent data that DC maturation in terms of production of cytokine and T cell cross-presentation is impaired. Yet the question still remains: why is immune response dampened with ferroptotic cancer cells? Authors discuss the potential role of PGE2 and oxPL, and indeed, lipid-laden DCs are known to result in loss of function. Are the authors able to observe lipid droplets in these DCs? Could they measure lipid species in ferroptotic cancer cell-loaded DCs? It would be of interest to identify the differences that define characteristics of initial vs intermediate/terminal.
2. It is intriguing that ferroptosis does not induce externalization of phosphatidylserine to the outer plasma membrane. What is the defining factor of phosphatidylserine-independent phagocytosis that is unique to terminal ferroptosis cancer cells and not the other stages?
3. In Figure 5, authors show hampered proliferation of CD8 T cells upon both pharmacological inhibition and genetic knockdown of Gpx4. As ferroptotic cancer cells do not successfully regress tumor growth, one can speculate that the effector response is also dampened entirely. However, could the authors also measure other forms of effector readouts such as IFN γ production, anti-tumor memory, CD4-help to assure that ferroptotic cancer cells are incapable of mounting anti-tumor response?
4. For the in vivo portion of this study, the authors provide % of tumor-free mice. Could the authors provide data on the tumor growth? Would the tumor growth in ferroptosis vaccinated condition still be similar to that of PBS control?
5. To induce necroptosis, the authors use the combination of TNF, TAK1i and zVAD-fmk. Indeed, TAK1 can regulate cell viability via RIPK1 pathway, but it can also regulate NK-kB activation. Could

the authors please discuss the usage of TAK1 as a necroptosis inducer and the effect it may have on NF- κ B signaling as well as downstream effector response in accordance to the paper by DOI: 10.1126/science.aad0395?

6. Extended Figure 1a/c: Why is there a difference in cell death/lipid peroxidation in ML162 vs RSL3? DFO does not restore cell survival and addition of Fer-1 does not abrogate lipid peroxidation. Do these results link to the non-specific nature of ML162 that is mentioned in Extended Fig 5b? Would the results be more conclusive with ML210?

Minor comments:

1. Figure legends do not indicate how many times the experiments were repeated.
2. The legend in Fig 3 includes explanation of the results. Please adjust accordingly to explain what had been performed in each subfigure instead.
3. For Figure 2b, please include expression patterns of necroptosis/apoptosis as a comparison to ferroptosis-induced DC maturation markers.
4. Figure 2e is missing a negative control (ie PBS).
5. The authors challenge previous findings by Efimova et al. yet use ML162 instead of RSL3 to justify their case (extended figure 3). Would the trajectory of death in prophylactic vaccination model using RSL3 still be similar to that of ML162 (this question arises due to the unspecificity of ML162)?

We wish to thank you and the reviewers for the insightful comments as they greatly helped us to improve the quality of our manuscript. Please find point by point responses to your remarks below:

NCOMMS-21-23582-T “Cancer cells dying from ferroptosis impede dendritic cell-mediated anti-tumor immunity” Wiernicki et al.	
Referee 1	
The manuscript “Cancer cells dying from ferroptosis impede dendritic cell mediated anti-tumor immunity” written by Bartosz Wiernicki and colleagues is an interesting paper that explores the capacity of ferroptosis to induce the so-called immunogenic cell death “ICD”. The major claim of the paper is ferroptosis does not induce ICD even though the production of DAMPS, and inflammatory cytokines. This conclusion is well supported by the data presented in the paper. The methodology is correct and well documented. Note that since I am not a statistician I did not evaluated the methods used for statistics. Strengths include the use an inducible model of ferroptosis relying on the doxycycline dependent knockdown of GPX4 allowing synchronized and complete cell death induction. The novelty of this study resides in the description of all the mechanisms usually associated with immunogenic cell death and yet ferroptosis does not induce immunogenicity. This suggests that the commonly studied markers associated with ICD (ATP, HMGB1, IFNβ...) may not be systematically correlated with ICD. This work is important since it challenges the mandatory role of DAMPS and other inflammatory cytokines for the induction of ICD.	We would like to thank this referee for the appreciation of the methodology, the conclusions, and the novelty of our study.
Calreticulin (CRT) exposure has been described as an important feature of type I ICD. Why didn't the authors monitor it? CRT should be monitored.	Following the reviewer’s advice, we performed flow cytometry experiments studying the presence of calreticulin during ferroptosis using three chemical ferroptosis inducers as well as the genetic model. The results show that in all instances, calreticulin is exposed during cell death, but in contrast to doxorubicin, at later stages of cell death just before the cells loose plasma membrane integrity. This resulted in changes in Figure 2c and additional Figures 1c, 1d, 2c and 2d and Extended Data Fig. 2b, 2c. The new results are described in lines 84-91, 121-124 and discussed in lines 284-291. Additionally, given that exposed calreticulin facilitates phagocytosis of dying cells (Feng et al., 2018), we analyzed whether blocking the membrane calreticulin can decrease phagocytosis of terminal ferroptotic cells. Our data revealed that blocking calreticulin exposure of terminally ferroptotic cells did not affect their phagocytosis by dendritic cells. The additional data also resulted in adaptations in the

	results (lines 171-172), discussion (lines 291-293) and the new Figure 3g.
A recent study published in 2020 seems to show contradictory data. Indeed, Luliia Efimova has published a study entitled “Vaccination with early ferroptotic cancer cells induces efficient antitumor immunity” J Immunother Cancer. 2020; 8(2): e001369. In this paper, authors used the same tumor cell line (MCA205) and demonstrated that early ferroptosis induced ICD contrary to late ferroptosis. Bartosz Wiernicki and colleagues discussed these results as “short treatment with ferroptosis inducers does not result in complete cell death induction and retains a high yield of living cells. Indeed, as we found and confirmed by the authors, removal of the chemical ferroptosis inducers after a short period of incubation does not result in full cell death.” This reflection is interesting but not necessarily convincing on several aspects.	We thank the reviewer for raising the issue of live cells component in the induction of immunogenic cell death, as it is often ignored in the field. The results of an ex-colleague of us (Dmitri Krysko) are indeed intriguing because with apparently opposite conclusion despite the use of an identical cancer model. We have tried to explain this opposite conclusion and think that the contribution of living cells at the vaccination site after short treatment with ferroptosis inducers as a model for “early ferroptosis” may explain the finding. This is why we have put so much effort to develop an inducible and synchronized model for ferroptosis, resulting in 100% cell death and no tumor growth at the vaccination site. By answering the items below raised by this referee, we hope that our reflections have become more convincing.
Firstly, when we use anti-cancer drugs in patients, death induction is never 100% synchronized and complete but still can favor immunogenicity. Authors should discuss this point	Our paper deals with understanding the effect of ferroptosis in a prophylactic vaccination model and trying to explain the absence of protective effect by further in vitro studies on dendritic cells. Patient treatment relying on partial cell death of tumor cells may indeed fuel immunogenic response. Tumors exposed to cytotoxic drugs can become senescent or undergo ‘cellular stress’, but can remain resistant to cell death and evoke cellular subroutines (e.g. production of cytokines) that can modulate immune system response. Whether ferroptosis inducers can cause senescence or cellular was not the scope of this study. Our initial attempts to study the impact of “early” (read partially killed cells i.e. still containing living cells) in the prophylactic vaccination model were met with failure, as it resulted in the growth of the tumor in vaccination site. According to the definition of immunogenic cell death based on prophylactic vaccination, tumor growth at the vaccination site should be absent to allow a conclusion on the immunogenicity of the cell death process (Galluzzi et al., 2020). In this adapted version we included these data in Extended data Figure 3b,c, and elaborated the findings in lines 103-107.
Secondly, this would imply that the living cells of the MCA205 model would have a vaccination effect. This was described for some tumor lines and published under the concept of concomitant immunity. This concomitant immunity is inhibited by immune-suppressive mechanisms such as regulatory T cells (Alan N. Houghton Group; J. Exp. Med. Volume 200, Number 6, 20 September 2004 771-782). However, this control (i.e. living MCA205 cells) is never included in experiments and has to be included in experiments.	To address the issue of concomitant immunity, we performed a prophylactic vaccination experiment analyzing the importance of live, tumorigenic MCA205 cells in the immunogenicity induction. Our data clearly demonstrates that the tumor growth can induce the immunogenicity as reported before in the publication mentioned by the reviewer and reported by our group (Aaes et al., 2016). Addition of ferroptotic cells during vaccination did not prevent the immunogenicity by the growing tumor. Therefore, we conclude that the MCA205-derived tumor can provide concomitant immunity, that is not

	affected by the initial presence of ferroptotic cells, probably because the effect of ferroptotic cells is fading out with time. Please see Extended Data Figure 3b,c.
Thirdly, as shown for mitoxantrone, about 50% of tumor cells are still alive (extended data, figure 1d) and yet the vaccine effect exists. Therefore, a question remains with mitoxantrone in authors' setting: is apoptotic cells or living cells inducing this vaccine effect?	Whether concomitant immunity contributes to the efficacy of Mitoxantrone-treated cells to provide immunological protection against challenge is possible, but unlikely. Concomitant immunity relies on continuous delivery of pro-immunogenic signals due to the tumor growth, an event that results only in minor cell death induction. Secondly, published data shows that the immunogenic efficacy of Mitoxantrone (and other anthracyclines) is diminished when cells are unable to expose calreticulin (Obeid et al., 2007) or to release HMGB1 (Yamazaki et al., 2014) regardless of the remaining non-permeabilized cells. Both events are characteristic for ICD and are not observed in untreated cells suggesting that further events associated with tumor growth are responsible for the induction of concomitant immunity. Noteworthy that we are dealing with highly immunogenic tumor models expressing retroviral antigens (Aaes et al. 2021). At the same time, our data show that the addition of ferroptotic cells to the vaccine, does not impact the concomitant immunogenicity of live cells as it does with the immunogenicity of MTX-induced cell death (Figure 6e). In this adapted version we included these data in Extended data Figure 3b,c and elaborated the findings in lines 103-107.
Thus, it is crucial to have the vaccine effect of living MCA205 cells to observe (or not) the induction of concomitant immunity. If so, this would imply that DAMPS and other inflammatory cytokines released by dying cells have an adjuvant effect only in the presence of concomitant immunity (induced by living tumor cells), therefore only with immunogenic tumors. It would be interesting to show by diluting living cells with increasing doses of ferroptosis-induced cell death that concomitant immunity is inhibited. The involvement of Treg in this mechanism should be also monitored. This may provide an explanation for the observation that ferroptosis is associated with poor prognosis in some patients (Zhu L, Cancer Cell Int (2021) 21(1):124; Zheng Y, J Cell Mol Med (2021) 25(6):3080–90; Jiang X, J Oncol (2021) 2021:6635526) even though ferroptosis could be correlated with immune infiltrate (Chen Ji. Front Oncol. 2021; 11: 700084).	Even if live cells (understood as cells 'destined' to die but not permeabilized at the moment of vaccination) may contribute to the immunogenicity, our data anyway shows that the injection of the initial ferroptotic iGPX4KD cells fails to establish immunological protection against challenge, contrary to the MTX-treated cells (Figure 2g). In both conditions no tumor growth was observed on the vaccination site. This setup allows to determine poor immunogenicity of ferroptosis regardless of the potential pro-immunogenic component of live cells. With regard to the involvement of Tregs in the in vivo action of ferroptotic cells, we think that this is a very interesting proposal for further research. Therefore, we have included this interesting point in the discussion section. This gives an interesting context of our experimental findings. In this adapted version we added information in line 299-300 and 318.
I have minor comments and questions:  1. Incubation time with cell death inducers should be clearly indicated for each experiments. 2. Rechallenge with only 30.000 MCA205 should be justified. What is the tumor uptake rate with this seemingly low cell count? 	 1. The necessary information was added to the figure legends. 2. The number of injected cells serving as a challenge in the prophylactic vaccination model complies with previously established protocols (Adjemian et al., 2020; Michaud et al., 2011;

 3. Figure 1b left panel, doxorubicin data are missing. Should be included. 4. Figure 2e and fig 6a and 6b: immunogenic apoptosis; necroptosis. No information of ICD inducer (apoptosis and necroptosis) is indicated in the figure legend 5. • Figure 4b: Does Cxcs5 stand for CXCR5? 	Michaud et al., 2014; Twitty et al., 2011), although we recognize that higher number of injected MCA205 also have been used. All the animals rechallenged with 30.000 cells “vaccinated” with PBS as negative control developed tumors. Tumorigenicity of the cell line has been established before and the range of the number of cells necessary to establish tumor growth after subcutaneous injection is lower (Korrer et al., 2014).  3. The suggested experiments were performed, and the data was added to the Figure 1b. 4. The information regarding the cell death induction details has been added to the figure legends (Fig. 2e is now Fig. 2g) 5. This has been corrected.
Referee 2	
Wiernicki and colleagues report that ferroptotic cancer therapy is not suitable for immunogenic cell death vaccination therapy as it negatively impacts the downstream immune response. It is a finding that depicts the unresolved immunogenic potential of ferroptotic cell death for prophylactic cancer cell vaccination. The paper is overall elegantly written and presented. Authors eloquently analyze the different stages of ferroptosis throughout the manuscript to identify the characteristics that are required for immunogenic cell death. The data is thorough as it not only compares pharmacological inhibition of Gpx4 by ML160 and RSL3 but also takes a genetic approach by inducible knockout of Gpx4. These systems together along with usage of inhibitor of SLC7A11, transporter for cysteine and ultimately glutathione building block strengthens the value of this study. There are some questions that do remain, however, and additional controls are also required. How ferroptosis induces immunogenic cell death profile (i.e. Release of DAMPs, cytokines, chemokines) yet experiences dampening of the adaptive immune response that is necessary for immunological protection is not well defined and insight here would be greatly appreciated. Provided the authors can address some of the questions below, this paper would be of great interest to the readership at Nature Communications.	We thank this referee for the appreciation of our work and the suggestions for further improvements and insights. Below we have tackled most of the items raised by this referee. However, we have made attempts for deeper in vivo analysis of the immune response, but we realized that further elaboration to come to conclusions based on decent data should be the subject of an additional extensive study. We hope that this referee will understand this decision. But having said this, we were able to address most of the questions raised by this referee.
 1. In figure 3b, authors illustrate expression of DC activation markers upon presentation with cancer cells at different ferroptotic stages, and show that exposure to cells at intermediate/terminal stages facilitate the upregulation of CD86, CD40 and MHC-II but not initial stage of ferroptosis. Although it is unclear whether the magnitudes of expression marker upregulation in intermediate/terminal stages are to the extent of apoptosis or necroptosis cancer cells, it is nonetheless coherent from the subsequent 	In the adapted version of the manuscript, we have documented the presence lipid droplets and recognize herewith the possible role of lipid accumulation in the dendritic cells and its negative impact on the functionality of antigen presenting cells, as it has been described before (Cao et al., 2014; Veglia et al., 2017). To address the potential accumulation of lipid droplets in the dendritic cells because of the phagocytosis of ferroptotic cells, we performed additional experiments relying on the

data that DC maturation in terms of production of cytokine and T cell cross-presentation is impaired. Yet the question still remains: why is immune response dampened with ferroptotic cancer cells? Authors discuss the potential role of PGE2 and oxPL, and indeed, lipid-laden DCs are known to result in loss of function. Are the authors able to observe lipid droplets in these DCs? Could they measure lipid species in ferroptotic cancer cell-loaded DCs? It would be of interest to identify the differences that define characteristics of initial vs intermediate/terminal.	BODIPY 493/503 nm probe, whose fluorescence reflects the level of lipid droplets accumulation. Our data demonstrate that co-culture with ferroptotic cells increases the level of lipid droplets in the dendritic cells. This increase was the strongest when dendritic cells were exposed to the initial stages of ferroptosis compared to the intermediate or terminal stage. The microscopy experiments further confirmed the high presence of lipid droplets in the dendritic cells. Moreover, accumulation of lipid droplets was higher when cells BMDC were incubated with ferroptotic compared to apoptotic or untreated cells. This resulted in additional Figures 3h,i, Extended data Figure 5c and additional explanation in Results (lines 172-182) and Discussion (lines 312-314).
2. It is intriguing that ferroptosis does not induce externalization of phosphatidylserine to the outer plasma membrane. What is the defining factor of phosphatidylserine-independent phagocytosis that is unique to terminal ferroptosis cancer cells and not the other stages?	Reflecting on that comment, we noticed that our statement in the manuscript was not precise enough. Indeed, we did not observe the PS exposure on the surface of ferroptotic cells, but that conclusion is valid only valid for cells that are not permeabilized. In the ruptured cells (“terminal ferroptotic cells”), it is impossible to determine whether the observed Annexin V signal increase reflects the intracellular or extracellular binding presence of Annexin-V-FITC to phosphatidylserine. For that reason, it cannot be excluded that ferroptotic cells in their terminal stage are engulfed in PS-dependent manner. To test that hypothesis, we performed the experiment where we pretreated the ferroptotic cells with Annexin V to block PS-dependent interaction during the incubation with dendritic cells. Our data showed that this treatment did not resolve in the changes in the uptake of the ferroptotic cells (unlike apoptotic cells), excluding the need for PS exposure in phagocytosis and suggesting other molecules to be involved, as has been described before for efferocytosis of apoptotic cells (Ucker et al., 2012). These data are mentioned in Figure 3f, Extended Data Figure 6d and lines 171-172. Related with this question on phagocytosis, we also included the possible role of calreticulin as a possible signal for recognition and phagocytosis. Data collected during the revision showed that calreticulin, an ‘eat me’ signal, is exposed during terminal stage of ferroptosis and therefore may be responsible for the uptake of terminally ferroptotic cells. To test that hypothesis, we pretreated ferroptotic targets with calreticulin antibody (or isotype control) and performed phagocytosis assay. We did not observe any changes in the uptake of ferroptotic cells, which suggests that ferroptotic cells are engulfed by other mechanism. These data are mentioned in Figure 3g and lines 171-172, and 291-293. Our observation that only terminally ferroptotic cells were phagocytosed was confirmed when we used macrophages, phagocytes that can engulf ferroptotic

	cells via interaction with oxidized phospholipids (Luo et al., 2021). These data are mentioned in the Extended Data Figure 6c. Overall, our data show that ferroptotic cells are engulfed via mechanisms that do not solely depend on PS or CRT and may include additional redundant mechanisms that remain to be identified.
3. In Figure 5, authors show hampered proliferation of CD8 T cells upon both pharmacological inhibition and genetic knockdown of Gpx4. As ferroptotic cancer cells do not successfully regress tumor growth, one can speculate that the effector response is also dampened entirely. However, could the authors also measure other forms of effector readouts such as IFNγ production, anti-tumor memory, CD4-help to assure that ferroptotic cancer cells are incapable of mounting anti-tumor response?	We acknowledge that providing additional information about in vivo the generation and the role of CD4 T cells following the interaction between dendritic cells and ferroptotic cells would be of great interest. Our attempts to perform experiments however did not produce results that would have high enough quality to be included in the manuscript. We would also like to put the emphasis on the fact that in our research we focused on the observed phenomenon of decreased antigenicity of ferroptosis which was confirmed by prophylactic vaccination with BM1-OVA and subsequent challenge with B16 melanoma cells expressing ovalbumin (Figure 6a) and therapeutic vaccination model relying on conventional dendritic cells engulfing ovalbumin-expressing ferroptotic and necroptotic cells (Figure 6c).
4. For the in vivo portion of this study, the authors provide % of tumor-free mice. Could the authors provide data on the tumor growth? Would the tumor growth in ferroptosis vaccinated condition still be similar to that of PBS control?	The graphs presenting the tumor size growth have been added to all the presented prophylactic vaccination model experiments. These data are mentioned in Figure 2h, Extended Data Figure 1e, Extended Data Figure 7c.
5. To induce necroptosis, the authors use the combination of TNF, TAK1i and zVAD-fmk. Indeed, TAK1 can regulate cell viability via RIPK1 pathway, but it can also regulate NF-κB activation. Could the authors please discuss the usage of TAK1 as a necroptosis inducer and the effect it may have on NF-κB signaling as well as downstream effector response in accordance to the paper by DOI: 10.1126/science.aad0395?	The work mentioned by the reviewer supports the notion that NF-κB is required for mounting successful protection against challenge (Yatim et al., 2015) while TAK1i can inhibit NF-κB activation (Bosman et al., 2014). The role of NF-κB in contributing to the immunogenicity of cell death has been controversial as other researchers showed that it is not absolutely required for the ICD induction (Aaes et al., 2016; Ren et al., 2017). We checked the NF-κB activation during ferroptosis. Our results clearly demonstrated that NF-κB is not activated during ferroptosis. These data are mentioned in Extended Data Figure 4d and lines 119-121.
6. Extended Figure 1a/c: Why is there a difference in cell death/lipid peroxidation in ML162 vs RSL3? DFO does not restore cell survival and addition of Fer-1 does not abrogate lipid peroxidation. Do these results link to the non-specific nature of ML162 that is mentioned in Extended Fig 5b? Would the results be more conclusive with ML210?	We do not know why in this instance we did not observe full rescue of cell survival after administration of iron chelator, but we cannot exclude the possibility that additional off-target activities of the drug may play a role. In order to establish the effectiveness of Fer1 in blocking lipid ROS accumulation, we performed additional experiments and used statistical analysis. Our data shows, that in all three ferroptosis inducers, Fer1 successfully blocks lipid peroxidation. These data are mentioned in Extended Data Figure 1c.
Minor comments:	

 1. Figure legends do not indicate how many times the experiments were repeated. 2. The legend in Fig 3 includes explanation of the results. Please adjust accordingly to explain what had been performed in each subfigure instead. 3. For Figure 2b, please include expression patterns of necroptosis/apoptosis as a comparison to ferroptosis-induced DC maturation markers. 4. Figure 2e is missing a negative control (ie PBS). 5. The authors challenge previous findings by Efimova et al. yet use ML162 instead of RSL3 to justify their case (extended figure 3). Would the trajectory of death in prophylactic vaccination model using RSL3 still be similar to that of ML162 (this question arises due to the unspecificity of ML162)? 	 1. We have included this information in the legends. 2. The legend in Figure 3 was adapted accordingly. 3. The comparison between different modes of cell death influencing the maturation of the dendritic cells is certainly research avenue worth exploring further. In our work however, we focused on establishing the role of ferroptotic cells, at different stages of cell death, in inducing the maturation of the dendritic cells. We determined that ferroptotic cells are able to induce maturation in terms of the exposure of co-stimulatory molecules and this fact cannot explain lack of immunogenicity after vaccination with ferroptotic cells. Our efforts focused on exploring the observed effect beyond the maturation of the antigen presenting cells and confirming inferiority of ferroptosis in relevant in vivo models. 4. The data coming from the negative control animals was pooled from all experiments and is presented in the Figure 1a. Now, data has been divided according to the experiment resulting in the change in Figure 1a and 2g. 5. We performed prophylactic vaccination experiment using RSL-3 as an inducer of ferroptosis. Our results show that RSL-3 – killed cells are not successful in mounting immunogenic response, similarly to ML162-induced cell death. These data are presented in Extended data Fig. 3b.
---	--

References

- Aaes, T. L., Kaczmarek, A., Delvaeye, T., Craene, B. D., Koker, S. D., Heyndrickx, L., Delrue, I., Taminau, J., Wiernicki, B., Groote, P. D., Garg, A. D., Leybaert, L., Grooten, J., Bertrand, M. J. M., Agostinis, P., Berx, G., Declercq, W., Vandenabeele, P. & Krysko, D. V. (2016). Vaccination with Necroptotic Cancer Cells Induces Efficient Anti-tumor Immunity. *Cell Reports*, *15*(2), 274–287. <https://doi.org/10.1016/j.celrep.2016.03.037>
- Aaes TL, Vandenabeele P. The intrinsic immunogenic properties of cancer cell lines, immunogenic cell death, and how these influence host antitumor immune responses. *Cell Death Differ*. 2021 Mar;*28*(3):843-860. doi: 10.1038/s41418-020-00658-y. Epub 2020 Nov 19. PMID: 33214663
- Adjemian, S., Oltean, T., Martens, S., Wiernicki, B., Goossens, V., Berghe, T. V., Cappe, B., Ladik, M., Riquet, F. B., Heyndrickx, L., Bridelance, J., Vuylsteke, M., Vandecasteele, K. & Vandenabeele, P. (2020). Ionizing radiation results in a mixture of cellular outcomes including mitotic catastrophe, senescence, methuosis, and iron-dependent cell death. *Cell Death & Disease*, *11*(11), 1003. <https://doi.org/10.1038/s41419-020-03209-y>

- Bosman, M. C. J., Schepers, H., Jaques, J., Brouwers-Vos, A. Z., Quax, W. J., Schuringa, J. J. & Vellenga, E. (2014). The TAK1-NF- κ B axis as therapeutic target for AML. *Blood*, *124*(20), 3130–3140. <https://doi.org/10.1182/blood-2014-04-569780>
- Cao, W., Ramakrishnan, R., Tuyrin, V. A., Veglia, F., Condamine, T., Amoscato, A., Mohammadyani, D., Johnson, J. J., Zhang, L. M., Klein-Seetharaman, J., Celis, E., Kagan, V. E. & Gabilovich, D. I. (2014). Oxidized Lipids Block Antigen Cross-Presentation by Dendritic Cells in Cancer. *The Journal of Immunology*, *192*(6), 2920–2931. <https://doi.org/10.4049/jimmunol.1302801>
- Feng, M., Marjon, K. D., Zhu, F., Weissman-Tsukamoto, R., Levett, A., Sullivan, K., Kao, K. S., Markovic, M., Bump, P. A., Jackson, H. M., Choi, T. S., Chen, J., Banuelos, A. M., Liu, J., Gip, P., Cheng, L., Wang, D. & Weissman, I. L. (2018). Programmed cell removal by calreticulin in tissue homeostasis and cancer. *Nature Communications*, *9*(1), 3194. <https://doi.org/10.1038/s41467-018-05211-7>
- Galluzzi, L., Vitale, I., Warren, S., Adjemian, S., Agostinis, P., Martinez, A. B., Chan, T. A., Coukos, G., Demaria, S., Deutsch, E., Draganov, D., Edelson, R. L., Formenti, S. C., Fucikova, J., Gabriele, L., Gaip, U. S., Gameiro, S. R., Garg, A. D., Golden, E., ... Marincola, F. M. (2020). Consensus guidelines for the definition, detection and interpretation of immunogenic cell death. *Journal for Immunotherapy of Cancer*, *8*(1), e000337. <https://doi.org/10.1136/jitc-2019-000337>
- Korrer, M. J., Zhang, Y. & Routes, J. M. (2014). Possible role of arginase-1 in concomitant tumor immunity. *PLoS One*, *9*(3), e91370. <https://doi.org/10.1371/journal.pone.0091370>
- Luo, X., Gong, H.-B., Gao, H.-Y., Wu, Y.-P., Sun, W.-Y., Li, Z.-Q., Wang, G., Liu, B., Liang, L., Kurihara, H., Duan, W.-J., Li, Y.-F. & He, R.-R. (2021). Oxygenated phosphatidylethanolamine navigates phagocytosis of ferroptotic cells by interacting with TLR2. *Cell Death & Differentiation*, 1–19. <https://doi.org/10.1038/s41418-020-00719-2>
- Michaud, M., Martins, I., Sukkurwala, A. Q., Adjemian, S., Ma, Y., Pellegatti, P., Shen, S., Kepp, O., Scoazec, M., Mignot, G., Rello-Varona, S., Tailler, M., Menger, L., Vacchelli, E., Galluzzi, L., Ghiringhelli, F., Virgilio, F. di, Zitvogel, L. & Kroemer, G. (2011). Autophagy-Dependent Anticancer Immune Responses Induced by Chemotherapeutic Agents in Mice. *Science (New York, N.Y.)*, *334*(6062), 1573–1577. <https://doi.org/10.1126/science.1208347>
- Michaud, M., Micaël, Sukkurwala, A. Q., Sano, F. D., Zitvogel, L., Kepp, O. & Kroemer, G. (2014). Synthetic induction of immunogenic cell death by genetic stimulation of endoplasmic reticulum stress. *Oncoimmunology*, *3*(3), e28276. <https://doi.org/10.4161/onci.28276>
- Obeid, M., Tesniere, A., Ghiringhelli, F., Fimia, G. M., Apetoh, L., Perfettini, J.-L., Castedo, M., Mignot, G., Panaretakis, T., Casares, N., Métivier, D., Larochette, N., Endert, P. van, Ciccocanti, F., Piacentini, M., Zitvogel, L. & Kroemer, G. (2007). Calreticulin exposure dictates the immunogenicity of cancer cell death. *Nature Medicine*, *13*(1), 54–61. <https://doi.org/10.1038/nm1523>
- Ren, J., Jia, X., Zhao, Y., Shi, W., Lu, J., Zhang, Y., Wu, J., Liang, B., Wu, R., Fu, G. & Han, J. (2017). The RIP3-RIP1-NF- κ B signaling axis is dispensable for necroptotic cells to elicit cross-priming of CD8(+) T cells. *Cellular and Molecular Immunology*, *14*(7), 639–642. <https://doi.org/10.1038/cmi.2017.31>
- Twitty, C. G., Jensen, S. M., Hu, H.-M. & Fox, B. A. (2011). Tumor-Derived Autophagosome Vaccine: Induction of Cross-Protective Immune Responses against Short-lived Proteins through a p62-

Dependent Mechanism. *Clinical Cancer Research*, 17(20), 6467–6481.
<https://doi.org/10.1158/1078-0432.ccr-11-0812>

Ucker, D. S., Jain, M. R., Pattabiraman, G., Palasiewicz, K., Birge, R. B. & Li, H. (2012). Externalized Glycolytic Enzymes Are Novel, Conserved, and Early Biomarkers of Apoptosis*. *Journal of Biological Chemistry*, 287(13), 10325–10343. <https://doi.org/10.1074/jbc.m111.314971>

Veglia, F., Tyurin, V. A., Mohammadyani, D., Blasi, M., Duperret, E. K., Donthireddy, L., Hashimoto, A., Kapralov, A., Amoscato, A., Angelini, R., Patel, S., Alicea-Torres, K., Weiner, D., Murphy, M. E., Klein-Seetharaman, J., Celis, E., Kagan, V. E. & Gabrilovich, D. I. (2017). Lipid bodies containing oxidatively truncated lipids block antigen cross-presentation by dendritic cells in cancer. *Nature Communications*, 8(1), 1–16. <https://doi.org/10.1038/s41467-017-02186-9>

Yamazaki, T., Hannani, D., Poirier-Colame, V., Ladoire, S., Locher, C., Sistigu, A., Prada, N., Adjemian, S., Catani, J. P. P., Freudenberg, M., Galanos, C., André, F., Kroemer, G. & Zitvogel, L. (2014). Defective immunogenic cell death of HMGB1-deficient tumors: compensatory therapy with TLR4 agonists. *Cell Death and Differentiation*, 21(1), 69–78. <https://doi.org/10.1038/cdd.2013.72>

Yatim, N., Jusforgues-Saklani, H., Orozco, S., Schulz, O., Silva, R. B. da, Sousa, C. R. E., Green, D. R., Oberst, A. & Albert, M. L. (2015). RIPK1 and NF- κ B signaling in dying cells determines cross-priming of CD8+ T cells. *Science (New York, N.Y.)*, 350(6258), 328–334.
<https://doi.org/10.1126/science.aad0395>

REVIEWERS' COMMENTS

Reviewer #1 (Remarks to the Author):

All the questions raised in the first review were answered with additional experiments, the results of which are convincing and demonstrate that ferroptosis is not immunogenic. Furthermore, the discussion of these points is clear and completes this work, which seems important because most of the markers currently monitored such as CRT, HMGB1... in patients may be present without being associated with ICD. This point should be highlighted in order to sensitize the medical community who determine the immunogenicity of an anti-cancer agent by such biomarkers. In the era of immunotherapy where chemotherapy/anti-PD-1 combinations are being tested, the addition of markers specific for ferroptosis, if any, and not only classical ICD markers as initially described by Kroemer's team, could be essential to avoid clinically ineffective combination therapies.

Reviewer #2 (Remarks to the Author):

The authors were able to answer the concerns that were previously raised. This study was very elegantly performed in any case and the revised version is very suited for publication.